# Learning Complete Protein Representation by Deep Coupling of Sequence and Structure

## Abstract

Learning effective representations is crucial for understanding proteins and their biological functions. Recent advancements in language models and graph neural networks have enabled protein models to leverage primary or tertiary structure information to learn representations. However, the lack of practical methods to deeply co-model the relationships between protein sequences and structures has led to suboptimal embeddings. In this work, we propose CoupleNet, a network that couples protein sequence and structure to obtain informative protein representations. CoupleNet incorporates multiple levels of features in proteins, including the residue identities and positions for sequences, as well as geometric representations for tertiary structures. We construct two types of graphs to model the extracted sequential features and structural geometries, achieving completeness on these graphs, respectively, and perform convolution on nodes and edges simultaneously to obtain superior embeddings. Experimental results on a range of tasks, such as protein fold classification and function prediction, demonstrate that our proposed model outperforms the state-of-the-art methods by large margins.

## 1  Introduction

Proteins are the fundamental building blocks of life and play essential roles in a diversity of applications, from therapeutics to materials. They are composed of 20 different basic amino acids, which are lined by peptide bonds and form a sequence. The one-dimensional (1D) sequence of a protein determines its structure, which in turn determines its biochemical function [40]. Due to recent progress in protein sequencing [34], massive numbers of protein sequences are now available. For example, the UniProt [3] database contains over 200 million protein sequences with annotations, *e.g.*, gene ontology (GO) terms, similar proteins, family and domains. Notably, the development of large-scale language models (LMs) in natural language processing has substantially benefited protein research owing to similarities between human language and protein sequences [16, 27]. For instance, models like ProtTrans [14] and ESM-series [39, 33] in learning protein representations have proven successful utility of pre-training protein LMs with self-supervision to process protein sequences.

Thanks to the recent significant progress made by AlphaFold2 [30] in three-dimensional (3D) structure prediction, a large number of protein structures from their sequence data are now made available. The latest release of AlphaFold protein structure database [43] provides broad coverage of UniProt [3]. Recently proposed structure-based protein encoders become to utilize geometric features [25, 24, 53], *e.g.*, ProNet [47] learns representations of proteins with 3D structures at different levels, like the amino acid, backbone or all-atom levels. There also exists a group of methods that build graph neural networks and LMs (LSTMs or attention models) to process sequence and structure [53, 50, 19], for example, GearNet [53] encodes sequential and spatial features by alternating node and edge message passing on protein residue graphs.

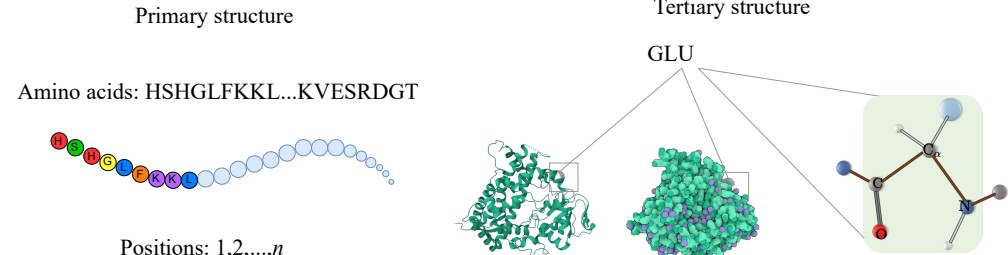

Figure 1: Illustration of the protein sequence and structure. 1) The primary structure comprises $n$ amino acids. 2) The tertiary structure with atom arrangement in Euclidean space is presented, where each atom has a specific 3D coordinate. Amino acids have fixed backbone atoms $(C_\alpha, C, N, O)$ and side-chain atoms that vary depending on the residue types. GLU: Glutamic acid. Complete geometries can be obtained based on these coordinates. The sequence and structure provide different information types and data categories.

The 1D sequence and 3D structure of a protein provide different types of information, in detail, as shown in Figure 1, compared with the 1D sequential order and amino acids in peptide chains, the tertiary structure provides 3D coordinates of each atom in protein residues, which allow them to perform precise functions. Although a protein's sequence determines its structure, various works have demonstrated the effectiveness of learning from either sequence or structure [33, 25]. However, rich constraints between the sequence and structure of a protein, which may be critical for protein tasks [4], have yet to be fully explored. Most protein sequence-structure modeling methods cannot deeply integrate the information behind sequence and structure for the reason that they tend to fuse representations together, extracted from sequence and structure encoders, respectively, by message passing mechanism [8] or by simple concatenation operations.

In this work, we aim to learn protein representations by deeply coupling the protein sequences and structures. Considering the relative positions of residues in the sequence and the spatial arrangement of atoms in the Euclidean space, the proposed CoupleNet constructs two categories of graphs for them, respectively. The complete representations are obtained at the amino acid and backbone levels on the two graphs, which are used as node and edge features to learn the final graph-level representations. Rather than concatenating sequence and structure representations, we take advantage of graph convolutions, performing node and edge convolutions simultaneously. The contributions of this paper are threefold:

- We propose a novel two-graph-based approach for representing the sequence and the 3D geometric structure of a protein, which is an effective way to guarantee completeness.

- We propose CoupleNet, a model that performs convolutions on nodes and edges of graphs to effectively integrate protein sequence and structure. This can better model the node-edge relationships and utilize the intrinsic associations between sequences and structures.

- Practically, the proposed model is verified by obtaining new state-of-the-art experimental results compared with current mainstream protein representation learning methods on a range of tasks, including protein fold classification, enzyme reaction classification, GO term prediction, domain prediction, and enzyme commission number prediction.

## 2  Related Work

**Protein Representation Learning**  Protein representation learning has become an active and promising direction in biology, which is essential to various downstream tasks in protein science. Because of the different levels of protein structures, existing methods mainly fall into three categories: protein LMs for sequences, structure models for geometry, and hybrid methods for both of them. As proteins are sequences of amino acids, considering their similarities with human languages, UniRep [1], UDSMProt [42] and SeqVec [23] use LSTM or its variants to learn sequence representations and long-range dependencies. TAPE [37] benchmarks a group of protein models, *e.g.*, 1D CNN, LSTM, and Transformer by various tasks. Elnaggar *et al.* [14] have trained six successful transformer variants

on billions of amino acid sequences, like ProtBert, and ProtT5. Similarly, ESM-series [39, 38, 33] employs a transformer architecture and a masked language modeling strategy to train robust representations based on large-scale databases. Besides the protein sequence, as we have stated before, the 3D geometric structure is vital to enhance protein representations. Most methods commonly seek to encode the spatial information of protein structures by convolutional neural networks (CNNs) [11], or graph neural networks [19, 2, 29]. For instance, SPROF [7] employs distance maps to predict protein sequence profiles, and IEConv [25] introduces a convolution operator to capture all relevant structural levels of a protein. GVP-GNN [29] designs the geometric vector perceptrons (GVP) for learning both scalar and vector features in an equivariant and invariant manner, Guo *et al.* [21] adopt SE(3)-invariant features as the model inputs and reconstruct gradients over 3D coordinates to avoid the usage of complicated SE(3)-equivariant models. ProNet [47] learns hierarchical protein representations at multiple tertiary structure levels of granularity. Moreover, CDConv [15] proposes continuous-discrete convolution using irregular and regular approaches to model the geometry and sequence structures. Some protein learning methods model the multi-level of structures at the same time [53, 6, 15], except for the primary structure and the tertiary structure, the second refers to the 3D form of local segments of proteins (e.g., $\alpha$-helix, $\beta$-strand), the quaternary is a protein multimer comprising multiple polypeptides, for example, PromtProtein [48] adopts a prompt-guided multi-task learning strategy for different protein structures with specific pre-training tasks. While previous works have attempted to combine protein sequence and structure, we focus on profoundly integrating them by specifically designing two types of graphs respectively and conducting convolutions simultaneously to learn protein representations.

**Complete Message Passing Mechanism**  ComENet [46] proposes rotation angles and spherical coordinates to fulfil the global completeness of 3D information on molecular graphs. By incorporating these designed geometric representations into the message passing scheme [18], the complete representation for a whole 3D graph is eventually yielded [47]. Unlike these methods, we couple sequence and structure via corresponding graphs and different geometric representations to obtain completeness representations.

## 3 Method

### 3.1 Preliminaries

**Notations**  We represent a 3D graph as $G = (\mathcal{V}, \mathcal{E}, \mathcal{P})$, where $\mathcal{V} = \{v_i\}_{i=1,\dots,n}$ and $\mathcal{E} = \{\varepsilon_{ij}\}_{i,j=1,\dots,n}$ denote the vertex and edge sets with $n$ nodes in total, respectively, and $\mathcal{P} = \{P_i\}_{i=1,\dots,n}$ is the set of position matrices, where $P_i \in \mathbb{R}^{k_i \times 3}$ represents the position matrix for node $v_i$. We treat each amino acid as a graph node for a protein, then $k_i$ depends on the number of atoms in the $i$-th amino acid. The node feature matrix is $X = [\boldsymbol{x}_i]_{i=1,\dots,n}$, where $\boldsymbol{x}_i \in \mathbb{R}^{d_v}$ is the feature vector of node $v_i$. The edge feature matrix is $E = [\boldsymbol{e}_{ij}]_{i,j=1,\dots,n}$, where $\boldsymbol{e}_{ij} \in \mathbb{R}^{d_\varepsilon}$ is the feature vector of edge $\varepsilon_{ij}$. $d_v$ and $d_\varepsilon$ denote the dimensions of feature vectors $\boldsymbol{x}_i$ and $\boldsymbol{e}_{ij}$.

**Invariance and Equivariance**  We consider affine transformations that preserve the distance between any two points, *i.e.*, the isometric group SE(3) in the Euclidean space. This is called the symmetry group, and it turns out that SE(3) is the special Euclidean group that includes 3D translations and the 3D rotation group SO(3) [17, 12]. The matrix form of SE(3) is provided in Appendix A.1.

Given the function $f : \mathbb{R}^m \to \mathbb{R}^{m'}$, assuming the given symmetry group $G$ acts on $\mathbb{R}^m$ and $\mathbb{R}^{m'}$, then $f$ is G-equivariant if,

$$f(T_g \boldsymbol{x}) = S_g f(\boldsymbol{x}), \; \forall \boldsymbol{x} \in \mathbb{R}^m, g \in G \tag{1}$$

where $T_g$ and $S_g$ are the transformations. For the SE(3) group, when $m' = 1$, the output of $f$ is a scalar, we have

$$f(T_g \boldsymbol{x}) = f(\boldsymbol{x}), \; \forall \boldsymbol{x} \in \mathbb{R}^m, g \in G \tag{2}$$

thus $f$ is SE(3)-invariant.

**Complete Geometric Representations** A geometric transformation $\mathcal{F}(\cdot)$ is complete if two 3D graphs $G^1 = (\mathcal{V}, \mathcal{E}, \mathcal{P}^1)$ and $G^2 = (\mathcal{V}, \mathcal{E}, \mathcal{P}^2)$, there exists $T_g \in \mathrm{SE}(3)$ such that the representations

$$\mathcal{F}(G^1) = \mathcal{F}(G^2) \Longleftrightarrow P_i^1 = T_g(P_i^2), \text{ for } i = 1, \dots n \tag{3}$$

The operation $T_g$ would not change the 3D conformation of a 3D graph [46]. Positions can generate geometric representations, which can also be recovered from them.

**Message Passing Paradigm** Message passing mechanism is mainly applied in graph convolutional networks (GCNs) [32], which follows an iterative scheme of updating node representations based on the feature aggregation from nearby nodes.

$$\begin{aligned}
\boldsymbol{h}_i^{(0)} &= \mathrm{BN}\left(\mathrm{FC}\left(\boldsymbol{x}_i\right)\right), \\
\boldsymbol{u}_i^{(l)} &= f_{\mathrm{Agg}}^{(l)}(\boldsymbol{h}_j^{(l-1)} | v_j \in \mathcal{N}(v_i)), \\
\boldsymbol{h}_i^{(l)} &= f_{\mathrm{Update}}^{(l)}(\boldsymbol{h}_j^{(l-1)}, \boldsymbol{u}_i^{(l)})
\end{aligned} \tag{4}$$

where $\mathrm{FC}(\cdot)$ and $\mathrm{BN}(\cdot)$ mean the linear transformation and batch normalization respectively. $\mathcal{N}(v_i)$ denotes the neighbours of node $v_i$. $f_{\mathrm{Agg}}^{(l)}$ and $f_{\mathrm{Update}}^{(l)}$ are aggregation and transformation functions at the $l$-th layer, which are permutation invariant and equivariant of node representations.

## 3.2 Sequence-Structure Graph Construction

Specifically, we represent each amino acid as a node, considering the residue types and their positions $i = 1, 2, \cdots, n$ (See Figure 1) in the sequence, we define the sequential graph primarily on the sequence, if $\|i - j\| < l$, the edge $\varepsilon_{ij}$ exists, where $l$ is a hyperparameter. Besides the sequential graph, we predefine a radius $r$, and build the radius graph, and there exists an edge between node $v_i$ and $v_j$ if $\|P_{i,\mathrm{C}\alpha} - P_{j,\mathrm{C}\alpha}\| < r$, where $P_{i,\mathrm{C}\alpha}$ denotes the 3D position of $\mathrm{C}_\alpha$ in the $i$-th residue.

Firstly, we design a base approach called $\mathrm{CoupleNet_{aa}}$ that only uses the $\mathrm{C}_\alpha$ positions of the structures. Inspired by Ingraham *et al.* [28], we construct a local coordinate system (LCS) for each residue, as shown in Figure 2.

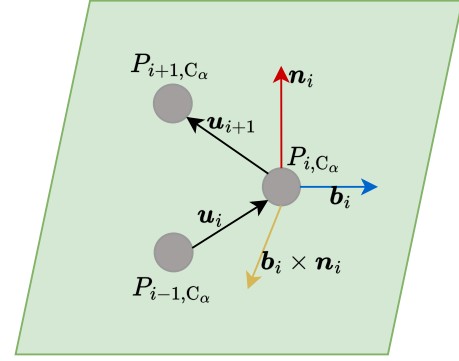

Figure 2: The local coordinate system.

$$\boldsymbol{Q}_i = [\boldsymbol{b_i} \quad \boldsymbol{n_i} \quad \boldsymbol{b_i} \times \boldsymbol{n_i}] \tag{5}$$

where $\boldsymbol{u}_i = \frac{P_{i,\mathrm{C}\alpha} - P_{i-1,\mathrm{C}\alpha}}{\|P_{i,\mathrm{C}\alpha} - P_{i-1,\mathrm{C}\alpha}\|}, \boldsymbol{b_i} = \frac{\boldsymbol{u}_i - \boldsymbol{u}_{i+1}}{\|\boldsymbol{u}_i - \boldsymbol{u}_{i+1}\|}, \boldsymbol{n_i} = \frac{\boldsymbol{u}_i \times \boldsymbol{u}_{i+1}}{\|\boldsymbol{u}_i \times \boldsymbol{u}_{i+1}\|}$. Then we can get the geometric representations at the amino acid level of a protein 3D graph,

$$\mathcal{F}(G)_{ij,aa} = \left(\|P_{i,\mathrm{C}\alpha} - P_{j,\mathrm{C}\alpha}\|, \boldsymbol{Q}_i^T \cdot \frac{P_{i,\mathrm{C}\alpha} - P_{j,\mathrm{C}\alpha}}{\|P_{i,\mathrm{C}\alpha} - P_{j,\mathrm{C}\alpha}\|}, \boldsymbol{Q}_i^T \cdot \boldsymbol{Q}_j\right) \tag{6}$$

where $\cdot$ is the matrix multiplication, this implementation is SE(3)-equivariant and obtains complete representations at the amino acid level; as if we have $\boldsymbol{Q}_i$, the LCS $\boldsymbol{Q}_j$ can be easily obtained by $\mathcal{F}(G)_{ij,aa}$.

For a node $v_i$, the node features $\boldsymbol{x}_{i,aa}$ in the baseline approach is the concatenation of the one-hot embeddings of the amino acid types and the physicochemical properties of each residue, namely, a steric parameter, hydrophobicity, volume, polarizability, isoelectric point, helix probability and sheet probability [51, 22], which provide quantitative insights into the biochemical nature of each amino acid. And $\mathcal{F}(G)_{ij,aa}$ is set as edge features for $\mathrm{CoupleNet_{aa}}$.

Secondly, we consider all backbone atoms $\mathrm{C}_\alpha, \mathrm{C}, \mathrm{N}, \mathrm{O}$ in CoupleNet. In detail, the peptide bond exhibits partial double-bond character due to resonance [20], indicating that the three non-hydrogen atoms comprising the bond (the carbonyl oxygen, carbonyl carbon, and amide nitrogen) are coplanar,

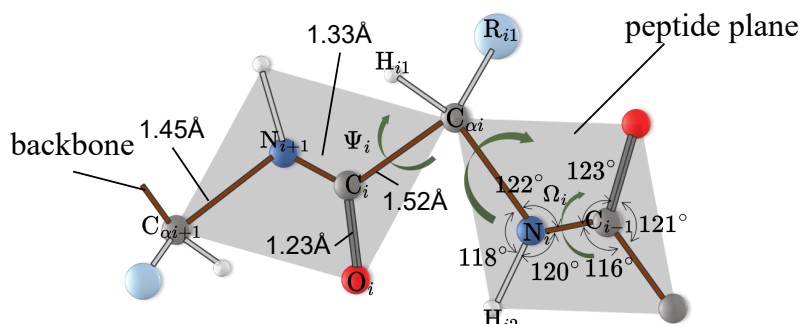

Figure 3: The polypeptide chain depicting the characteristic backbone bond lengths, angles, and torsion angles ($\Psi_i, \Phi_i, \Omega_i$). The planar peptide groups are denoted as shaded gray regions, indicating that the peptide plane differs from the geometric plane calculated based on the 3D positions.

as shown in Figure 3. There is some rotation about the connection. The $N_i - C_{\alpha i}$ and $C_{\alpha i} - C_i$ bonds, are the two bonds in the basic repeating unit of the polypeptide backbone. These single bonds allow unrestricted rotation until sterically restricted by side chains [35, 45]. Since the coordinates of $C_\alpha$ can be obtained as we have the complete representations at the amino acid level, the coordinates of other backbone atoms based on these rigid bond lengths and angles are able to be determined with the remaining degree of the backbone torsion angles $\Phi_i, \Psi_i, \Omega_i$. The omega torsion angle around the $C - N$ peptide bond is typically restricted to nearly $180°$ (trans) but can approach $0°$ (cis) in rare instances. Other than the bond lengths and angles presented in Figure 3, all the H bond lengths measure approximately 1 Å.

For the sequential graph, we compute the sine and cosine values of $\Phi_i, \Psi_i, \Omega_i$ for each amino acid $i$, and use them as another part of nodes features for node $v_i$.

$$\boldsymbol{x}_i = \boldsymbol{x}_{i,aa} \| ((\sin \wedge \cos)(\Phi_i, \Psi_i, \Omega_i)) \tag{7}$$

where $\|$ denotes concatenation. There is no isolated node for the designed graph, which means the backbone atoms can be determined one by one along the polypeptide chain based on the positions of $C_\alpha$ and these three backbone dihedral angles. Therefore, the existing presentations $[\mathcal{F}(G)_{ij,aa}]_{i,j=1,...,n}$ and $[\boldsymbol{x}_i]_{i=1,...,n}$ are complete at the backbone level for the sequential graph.

For the radius graph, we want to get the positions of backbone atoms in any two amino acids $i$ and $j$. Inspired by trRosetta [52], the relative rotation and distance are computed including the distance ($d_{ij,C_\beta}$), three dihedral angles ($\omega_{ij}, \theta_{ij}, \theta_{ji}$) and two planar angles ($\varphi_{ij}, \varphi_{ji}$), as shown in Figure 4, where $d_{ij,C_\beta} = d_{ji,C_\beta}, \omega_{ij} = \omega_{ji}$, but $\theta$ and $\varphi$ values depend on the order of residues. These interresidue geometries define the relative locations of the backbone atoms of two residues in all their details [52], because the torsion angles of $N_i - C_{\alpha i}$ and $C_{\alpha i} - C_i$ do not influence their positions. Therefore, these six geometries are complete for amino acids at the backbone level for the radius graph. The graph edges contain the relative spatial information between any two neighboring amino acids $\boldsymbol{e}_{ij} = \mathcal{F}(G)_{ij,aa} \| \mathcal{F}(G)_{ij,bb}$, where

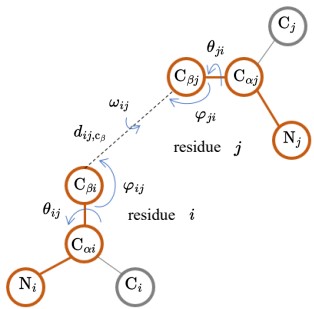

Figure 4: Interresidue geometries including angles and distances.

$$\mathcal{F}(G)_{ij,bb} = (d_{ij,C_\beta}, (\sin \wedge \cos)(\omega_{ij}, \theta_{ij}, \varphi_{ij})) \tag{8}$$

The designed node and edge features, $\boldsymbol{x}_i$ and $\boldsymbol{e}_{ij}$, for the sequential and radius graphs, provide a new perspective to represent protein sequences and structures. Such integration can bring better performance for the following graph-based learning tasks.

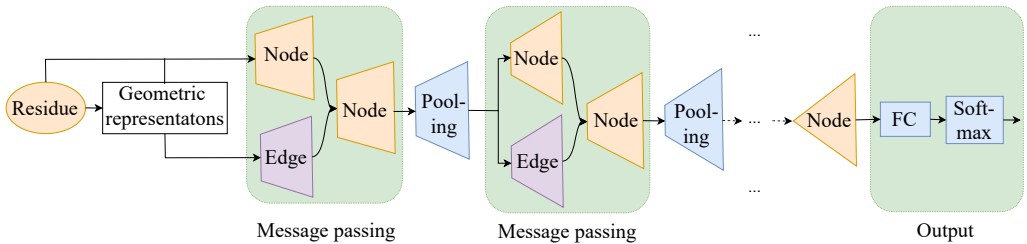

Figure 5: An illustration of CoupleNet.

## 3.3 Secqunce-Structure Graph Convolution

Inspired by the message passing paradigm and continuous-discrete convolution [15], sequences and structures are encoded successfully together by convolutions. To deeply couple sequences and structures of proteins and encode them jointly, we employ convolution to embed them simultaneously, exploring their relationships to generate comprehensive and effective embeddings. Different from previous works, we innovatively construct two categories of graphs for sequence and structure and design various sequential and structural representations to achieve completeness on them at the amino acid and backbone levels. We then convolve node and edge features with the help of the message passing mechanism.

In order to implement convolution on nodes and edges simultaneously between sequence and structure, we set $\varepsilon_{ij}$ to exist if the following conditions are satisfied

$$\|i - j\| < l \quad \text{and} \quad \|P_{i,\text{C}\alpha} - P_{j,\text{C}\alpha}\| < r \tag{9}$$

The existing node and edge feature matrices $(X, E)$ are complete representations of a protein 3D graph to reconstruct its backbone atom positions. Compared with the equation Eq. 4, the proposed CoupleNet first apply a $\text{FC}(\cdot)$ layer and a $\text{BN}(\cdot)$ layer to the node features to obtain the initial encoded representation. Then the $f_{\text{Agg}}^{(l)}$ is applied to gather neighboring features of nodes and edges by convolution, where $\sigma(\cdot)$ is the activation function. We use the dropout and add a residual connection from the previous layer as $f_{\text{Update}}^{(l)}$. For the consideration that the spatial arrangement and tight positioning of specific amino acids, which may be spaced widely apart on the linear polypeptide, are necessary for proteins to operate as intended [10], $l$ is set to be a relatively large number, see Appendix A.2 for details.

$$
\begin{aligned}
\boldsymbol{h}_i^{(0)} &= \text{BN}\left(\text{FC}\left(\boldsymbol{x}_i\right)\right), \\
\boldsymbol{u}_i^{(l)} &= \sigma(\text{BN}(\sum_{v_j \in \mathcal{N}(v_i)} W\boldsymbol{e}_{ij}\boldsymbol{h}_j^{(l-1)})), \\
\boldsymbol{h}_i^{(l)} &= \boldsymbol{h}_i^{(l)} + \text{Dropout}(\boldsymbol{u}_i^{(l)})
\end{aligned}
\tag{10}
$$

## 3.4 Model Architecture

Building upon the sequence-structure graph convolution, we build the CoupleNet, as shown in Figure 5. The inputs to the graph are the calculated sequential and structural representations $(X, E)$. Following the existing protein graph models [15, 25, 47], our CoupleNet employs graph pooling layers to obtain deeply encoded, graph-level representations. After pooling, due to the decrease in nodes, we increase the predefined radius $r$ to include more neighbors. The message passing mechanism only executes on nodes for the consideration of reducing model complexity. Another reason is that representations of sequences and structures have already been coupled by equation Eq. 4. A detailed description of the model architecture is provided in Appendix A.2.

Table 1: Accuracy (%) on fold classification and enzyme reaction classification. [*] means the results are taken from [15]. The best and suboptimal results are shown in bold and underline.

| Input | Method | Fold Classification | | | Enzyme |
|---|---|---|---|---|---|
| | | Fold | SuperFamily | Family | Reaction |
| Sequence | CNN [41]* | 11.3 | 13.4 | 53.4 | 51.7 |
| | ResNet [37]* | 10.1 | 7.21 | 23.5 | 24.1 |
| | LSTM [37]* | 6.41 | 4.33 | 18.1 | 11.0 |
| | Transformer [37]* | 9.22 | 8.81 | 40.4 | 26.6 |
| Structure | GCN [32]* | 16.8 | 21.3 | 82.8 | 67.3 |
| | GAT [44]* | 12.4 | 16.5 | 72.7 | 55.6 |
| | 3DCNN_MQA [11]* | 31.6 | 45.4 | 92.5 | 72.2 |
| | IEConv (atom level) [25]* | 45.0 | 69.7 | 98.9 | 87.2 |
| Sequence-Structure | GraphQA [2]* | 23.7 | 32.5 | 84.4 | 60.8 |
| | GVP [29]* | 16.0 | 22.5 | 83.8 | 65.5 |
| | ProNet-Amino Acid [47] | 51.5 | 69.9 | 99.0 | 86.0 |
| | ProNet-Backbone [47] | 52.7 | 70.3 | 99.3 | 86.4 |
| | ProNet-All-Atom [47] | 52.1 | 69.0 | 99.0 | 85.6 |
| | IEConv (residue level) [25]* | 47.6 | 70.2 | 99.2 | 87.2 |
| | GearNet [53] | 28.4 | 42.6 | 95.3 | 79.4 |
| | GearNet-IEConv [53] | 42.3 | 64.1 | 99.1 | 83.7 |
| | GearNet-Edge [53] | 44.0 | 66.7 | 99.1 | 86.6 |
| | GearNet-Edge-IEConv [53] | 48.3 | 70.3 | 99.5 | 85.3 |
| | CDConv [15] | 56.7 | 77.7 | 99.6 | 88.5 |
| | CoupleNet (Proposed) | **60.6** | **82.1** | **99.7** | **89.0** |

# 4 Experiments

## 4.1 Datasets and Settings

The models are trained with the Adam optimizer [31] using the PyTorch and PyTorch Geometric libraries. Detailed descriptions of the datasets and experimental settings are provided in Appendix A.3. Following the tasks in IEconv [25], GearNet [53] and CDConv [15], here, we evaluate the CoupleNet on four protein tasks: protein fold classification, enzyme reaction classification, GO term prediction and enzyme commission (EC) number prediction.

**Fold Classification**    Protein fold is to predict the fold class label given a protein, which is crucial for understanding how protein structure and protein evolution interact [26]. In total, this dataset contains $16,712$ proteins with $1,195$ fold classes. There are three test sets available, Fold: Training excludes proteins from the same superfamily. Superfamily: Training does not include proteins from the same family. Family: Proteins from the same family are included in the training.

**Enzyme Reaction Classification**    Reaction categorization aims to predict a protein's class of enzyme-catalyzed reactions, according to all four levels of the EC number [49, 36]. Following the setting in [25], this dataset has $37,248$ proteins from 384 four-level EC numbers [5].

**GO Term Prediction**    The goal of GO term prediction is to foretell whether a protein is related to a certain GO term. Following [19], these proteins are organized into three ontologies: molecular function (MF), biological process (BP), and cellular component (CC), which are hierarchically connected, functional classes. MF describes activities that occur at the molecular level, BP represents the larger processes, and CC describes the parts of a cell or its extracellular environment [3].

**EC Number Prediction**    This task seeks to predict the 538 EC numbers from the third level and fourth levels of different proteins [19], which describe their catalysis of biochemical reactions.

Table 2: $F_{max}$ on GO term and EC number prediction. [*] means the results are taken from [15]. The best and suboptimal results are shown in bold and underline.

| Category | Method | GO-BP | GO-MF | GO-CC | EC |
|---|---|---|---|---|---|
| Sequence | CNN [41]* | 0.244 | 0.354 | 0.287 | 0.545 |
| | ResNet [37]* | 0.280 | 0.405 | 0.304 | 0.605 |
| | LSTM [37]* | 0.225 | 0.321 | 0.283 | 0.425 |
| | Transformer [37]* | 0.264 | 0.211 | 0.405 | 0.238 |
| Structure | GCN [32]* | 0.252 | 0.195 | 0.329 | 0.320 |
| | GAT [44]* | 0.284 | 0.317 | 0.385 | 0.368 |
| | 3DCNN_MQA [11]* | 0.240 | 0.147 | 0.305 | 0.077 |
| Sequence-Structure | GraphQA [2]* | 0.308 | 0.329 | 0.413 | 0.509 |
| | GVP [29]* | 0.326 | 0.426 | 0.420 | 0.489 |
| | IEConv (residue level) [25]* | 0.421 | 0.624 | 0.431 | - |
| | GearNet [53] | 0.356 | 0.503 | 0.414 | 0.730 |
| | GearNet-IEConv [53] | 0.381 | 0.563 | 0.422 | 0.800 |
| | GearNet-Edge [53] | 0.403 | 0.580 | 0.450 | 0.810 |
| | GearNet-Edge-IEConv [53] | 0.400 | 0.581 | 0.430 | 0.810 |
| | CDConv [15] | 0.453 | 0.654 | 0.479 | 0.820 |
| | CoupleNet (Proposed) | **0.467** | **0.669** | **0.494** | **0.866** |

## 4.2 Baselines

We compare our proposed method with existing protein representation learning methods, which are classified into three categories based on their inputs, which could be a sequence (amino acid sequence), 3D structure or both sequence and structure. 1) Sequence-based encoders, including CNN [41], ResNet [37], LSTM [37] and Transformer [37]. 2) Structure-based methods (GCN [32], GAT [44], 3DCNN_MQA [11], IEConv (atom level) [25]). 3) Sequence-structure based models, *e.g.*, GVP [29], ProNet [47], GearNet [53], CDConv [15], *etc*. GearNet-IEConv and GearNetEdge-IEConv [53] add the IEConv layer based on GearNet, which is found important in fold classification.

## 4.3 Resluts of Fold and Reaction Classification.

Table 1 provides the comparisons on the fold and enzyme reaction classification. The results are reported in terms of accuracy (%) for these two tasks. From this table, we can see that the proposed model CoupleNet achieves the best performance across all four test sets on the fold and enzyme reaction classification compared with recent state-of-the-art methods. Especially on the Fold and SuperFamily test sets, CoupleNet improves the results by about 4%, showing that CoupleNet is proficient at learning the mapping between protein sequences, structures and functions. Moreover, CDConv [15] ranks second among these methods, both CDConv and our method are implemented by sequence-structure convolution. This phenomenon illustrates that deeply coupling sequences and structures of proteins is conducive to learning better protein embeddings. And our proposed CoupleNet model utilizes complete geometric representations and the specially designed message passing mechanism, achieving new state-of-the-art results.

## 4.4 Results of GO Term and EC Prediction

We follow the split method in [19, 53] to guarantee that the test set only comprises PDB chains with sequence identity no higher than 95% to the training set for GO term and EC number prediction. Table 2 compares different protein modeling methods on GO term prediction and EC number prediction. The results are reported in terms of $F_{max}$, which considers both precision and recall for evaluation, the equation of $F_{max}$ is provided in Appendix A.4. The proposed model, CoupleNet yields the highest $F_{max}$ across these four test sets of two tasks, outperforming other state-of-the-art models. This indicates CoupleNet can effectively predict the functions, locations, and enzymatic activities of proteins. These results once again illustrate the importance of deeply coupled sequences

Table 3: Ablation of our proposed method

| Method | Fold Classification | | | Enzyme | GO | | | EC |
| --- | --- | --- | --- | --- | --- | --- | --- | --- |
| | Fold | Superfamily | Family | Reaction | BP | MF | CC | |
| CoupleNet | 60.6 | 82.1 | 99.7 | 89.0 | 0.467 | 0.669 | 0.494 | 0.866 |
| CoupleNet$_{aa}$ | 57.8 | 78.7 | 99.6 | 88.6 | 0.458 | 0.660 | 0.484 | 0.851 |
| w/o $\Phi, \Psi, \Omega$ | 60.3 | 81.3 | 99.6 | 88.7 | 0.463 | 0.666 | 0.490 | 0.862 |
| w/o $d, \omega, \theta, \varphi$ | 60.4 | 81.5 | 99.7 | 88.9 | 0.461 | 0.666 | 0.488 | 0.864 |

and structures. The improvements of CoupleNet over the suboptimal CDConv [15] model indicate the advanced modeling power of CoupleNet.

We employ different cutoff splits following [19, 15], which means that the proteins in the test set are divided into groups that have, respectively, 30%, 40%, 50%, 70%, and 95% similarity to the training set for GO term and EC number prediction, as shown in Figure 6 and Appendix A.5. From Figure 6, we can see that our proposed model CoupleNet achieves the highest $F_{max}$ scores across all cutoffs, especially when the cutoffs are at 30% to 50%. There is a larger margin compared with GearNet, GearNet-Edge [53] and CDConv [15]. This demonstrates that CoupleNet, which utilizes complete geometric representations, is more robust, especially when there is a low similarity between the training and test sets.

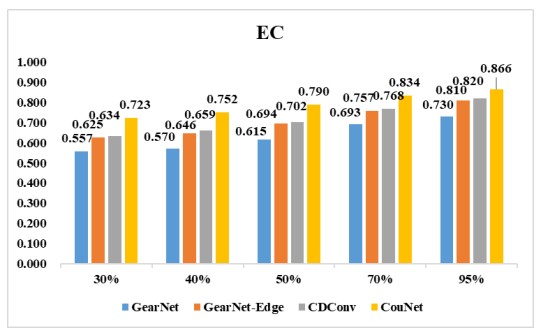

Figure 6: $F_{max}$ on EC number prediction under different cutoffs.

## 4.5 Ablation Study

Table 3 presents an ablation study of the proposed CoupleNet model on the four protein tasks. We examined the impact of removing the backbone torsion angles (w/o $\Phi, \Psi, \Omega$) and removing the interresidue geometric structure representations (w/o $d, \omega, \theta, \varphi$). The former is designed for the sequential graph, and the latter is for the radius graph to achieve completeness at the protein backbone level. However, we combine the two types of graphs together to enhance the relationships between sequence and structure. From Table 3, we can also find that these complete geometries provide complementary information to amino acid position features, with one of their removals leading to minor performance drops for the reason that they both provide complete geometries from different perspectives. Removing $\Phi, \Psi, \Omega$ causes larger performance degradation compared with removing $d, \omega, \theta, \varphi$. Such comparisons indicate that the backbone dihedral angles may have more effects on learning protein representations in these experimental settings. Compared with CoupleNet$_{aa}$, CoupleNet achieves significant improvements on the four tasks, demonstrating the importance of complete structural representations at the backbone level in learning protein embeddings.

## 5 Conclusions and Limitations

In this work, we propose CoupleNet, a novel protein representation learning method that deeply fuses protein sequences and multi-level structures by conducting convolution on graph nodes and edges simultaneously. We design the sequential graph and the radius graph, achieving completeness on them at different protein structure levels. Our approach achieves new state-of-the-art results on the protein tasks, which demonstrates the superiority our the proposed method. A limitation is that the detailed inter-relationships between sequence and structures remain to be explored and uncovered. We leave such research for future work.

While our model can enable advanced protein analyses and provide effective representations, there may exist broader impacts and harmful activities. The representations could potentially be misused, *e.g.*, for designing harmful molecules or proteins.

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
