# OpenReview forum: "Learning Complete Protein Representation by Deep Coupling of Sequence and Structure"
_NeurIPS.cc/2023/Conference — Submitted to NeurIPS 2023_

### Official Review · Reviewer_rGTX · 2023-07-02

**Soundness:** 3 good
**Presentation:** 3 good
**Contribution:** 2 fair
**Rating:** 4
**Confidence:** 3

**Summary:**

The paper addresses the challenge of integrating protein sequence and structure information to improve protein representations. The authors propose CoupleNet, a network that utilizes graph convolutions to model the relationships between protein sequences and structures. The approach involves constructing two types of graphs to model sequential features and structural geometries and performing convolutions on nodes and edges simultaneously. The proposed approach outperforms state-of-the-art methods on various protein-related tasks.

**Strengths:**

1) Comprehensive Protein Features: CoupleNet incorporates multiple levels of features in proteins, including residue identities, positions, and geometric representations. By considering both primary (sequence) and tertiary (structure) information, the model captures different aspects of proteins, leading to more informative and rich representations. The authors have demonstrated good work in engineering the features of their model. By carefully considering both primary (sequence) and tertiary (structure) information, the model has succeeded in capturing diverse aspects of proteins, leading to more informative and robust representations.

2) Experimental Results: The paper presents experimental results on a range of protein-related tasks, including protein fold classification, function prediction, and domain prediction. The results demonstrate that CoupleNet outperforms state-of-the-art methods by large margins. The comprehensive evaluation and superior performance validate the effectiveness of the proposed model.

**Weaknesses:**

1) Limited Novelty: While the proposed approach in CoupleNet is interesting and effective, it does not offer a significant departure from existing methods. Similar approaches, such as GearNet, have previously explored this concept by integrating the radius and sequence information as different edge types within a single graph. Therefore, the idea of constructing two separate graphs in CoupleNet, while slightly different in implementation, does not present a significant departure from existing methods.

2) A weakness of the paper is the lack of a detailed explanation regarding why the proposed model, CoupleNet, performs better than the state-of-the-art methods. While the experimental results demonstrate superior performance, the authors do not provide a thorough analysis or insights into the specific aspects of the model architecture or design choices that contribute to its improved performance. Without a clear explanation of the underlying factors that make the model more effective, it becomes challenging for readers to fully understand and interpret the advantages of CoupleNet over existing approaches.

3) Lack of Code Release: One notable weakness of the paper is the absence of code release. The lack of code availability hinders the reproducibility and transparency of the research.


[1] Zhang, Zuobai, et al. "Protein representation learning by geometric structure pretraining.

**Questions:**

1) Can you explain why the use of two separate graphs in CoupleNet is preferable to integrating sequence and structure information as different edge types within a single graph, as demonstrated in previous works like GearNet?

2) Could you clarify the computational complexity of the proposed model? How does the performance of CoupleNet scale with increasing data volume, particularly as protein databases continue to grow in size?




**Limitations:**

yes

---

> ### Author Rebuttal · Authors · 2023-08-10
>
> **Q1** Differences with GearNet.
>
> **A1** Thanks for the valuable feedback! (1) CoupleNet uses two graph types (radius and sequential) rather than GearNet's [1] three (adding k-nearest), the sequential edges in the sequential graph of GearNet are further divided into different edge types according to the relative position between two end nodes. Removing k-nearest avoids distant nodes. (2) CoupleNet adaptively adjust radius thresholds rather than using a fixed value. This can better capture different distance relationships. (3) CoupleNet incorporates more comprehensive feature information at the amino acid and backbone levels to help the network learn valuable patterns. (4) CoupleNet simultaneously convolves node and edge features together, while GearNet processes them separately.
> ***
> **Q2** Analyse the valuable parts.
>
> **A2** Thanks! (1) We combine some results from Table 1-3 in the manuscript:
>  | Method | Fold | SuperFamily | Family | Enzyme Reaction | GO-BP | GO-MF | GO-CC | EC |
> | :--- | :---: | :---: | :---: | :---: |  :---: |  :---: |  :---: |  :---: |
> |CDConv [2] | 56.7 | 77.7 | 99.6 | 88.5 | 0.453 | 0.654 | 0.479 | 0.820 |
> | CoupleNet | 60.6 | 82.1 | 99.7 | 89.0 | 0.467 | 0.669 | 0.494 | 0.886 |
> |$\mathrm{CoupleNet_{aa}}$| 57.8 |  78.7 |  99.6 | 88.6 | 0.458 | 0.660| 0.484 | 0.851 |
>
> In the above table, the input features of $\mathrm{CoupleNet_{aa}}$ and CDConv are similar, both operating at the amino acid level, using the positions of $\mathrm{C_\alpha}$. $\mathrm{CoupleNet_{aa}}$ obtains a little higher results than CDConv, which illustrates the effectiveness of our model design. (2) As shown in Table 6, removing the sequence information causes a slight performance decrease, which illustrates that the complete geometries provided in this paper are essential to achieve remarkable results.
> (3) From Table 3 in the manuscript, we can find that either removing the backbone torsion angles or removing the interresidue geometric structure representations leads to a minor performance drop, which may imply that these two groups of complete geometries provide similar information in terms of protein structures.
> ***
> **Q3** Code Release.
>
> **A3** Thanks! The code will be released upon the acceptance of this paper.
> ***
> **Q4** Why CoupleNet is preferable to integrating sequence and structure information.
>
> **A4** Thanks! (1) As we have stated before, the proposed model, CoupleNet, incorporates multiple levels and types of information in protein. These representations are fused together to be used as graph node and edge features. There are different edge types in GearNet, which implements message passing operations on nodes and edges separately for different types of graphs. We aim to implement convolution on nodes and edges simultaneously between protein sequences and structures; thus, the predefined sequential graph and radius graph are combined. This operation can be seen in Eq. 9 in the manuscripts. The combined graph has complete protein representations. (2) Another reason is that when the thresholds $r$ and $l$ are small, Eq. 9 defines the local environment [3] of an amino acid in the protein, which can reveal local dynamics and provide crucial context for a residue identity. In experiments, we conduct sequence pooling operations and enlarge the threshold of $r$ to contain distant or rare nodes, the constructions of relationships from local to global are more apparent.
> ***
> **Q5** Computational complexity Analysis.
>
> **A5** Thanks for your helpful feedback! (1) In the proposed CoupleNet, The sequential and radius graphs are combined via convolution (Eq. 9). For the computational complexity of the proposed model, considering one message passing layer, per-layer complexity is $\mathcal{O}(n\ast d_n )$ where $d_n$ is average node degree, much less than $n$. There are four average pooling layers, which can reduce the sequence length, but we enlarge the initial predefined radius, assuming the average degree of each node is still $d_n$. Using $B_s$ denotes the size of batch size; the final computational complexity is $\mathcal{O}(B_s n d_n )$. (2) When protein databases continue to grow in size, the computational complexity is linear with the number of residues and proteins. From Tabel 1 and Table 2 in the paper, we can see CoupleNet outperforms state-of-the-art methods by large margins. We get some results from [1] and [3], and compare them with some pretraining methods:
> | Method | Fold | SuperFamily | Family | Enzyme Reaction |
> | :--- | :---: | :---: | :---: | :---: |
> |ESM-1b | 26.8 | 60.1 | 97.8 | 83.1 |
> |ProtBERT-BFD| 26.6 | 55.8 | 97.6 | 72.2 |
> |GearNet-Edge-IEConv (Dihedral Prediction) | 51.8 | 77.8 | 99.6 | 87.0 |
> | CoupleNet | 60.6 | 82.1 | 99.7 | 89.0 |
>
> The above table shows comparisons of our model with a few current pertaining methods on the task of protein fold classification and enzyme catalytic reaction classification. Our proposed method can achieve comparable results even with pretraining methods. This indicates the proposed model may be more data-efficient, elicit more from limited information, and deal with more large-size databases.
> ***
> [1] Zuobai Zhang, et al. "Protein representation learning by geometric structure pretraining". In: International Conference on Learning Representations. 2023.
>
> [2] Hehe Fan, et al. "Continuous-Discrete Convolution for Geometry-Sequence Modeling in Proteins". In: The Eleventh International Conference on Learning Representations. 2023.
>
> [3] Huang B, et al. Accurate and efficient protein sequence design through learning concise local environment of residues[J]. Bioinformatics, 2023.
> ***
> Thank you again for all the efforts that help us improve our manuscript! In case our answers have justifiably addressed your concerns, we respectfully thank you that support the acceptance of our work. As you know, your support holds great significance for us. Also, please let us know if you have any further questions. Look forward to further discussions!

---

> > ### Comment · Reviewer_rGTX · 2023-08-17
> > **Response to Authors**
> >
> > Thank you for taking the time to provide a detailed rebuttal.
> > While I acknowledge your dedication to refining your manuscript and your insightful responses, I must express that I still have reservations about the novelty of the method and its claimed improvements compared to GearNet and the other baselines. Specifically, concerning the ability of the proposed model to capture long-range dependencies, there are plenty of recent GNN methods that have demonstrated capabilities in handling such dependencies effectively such as [1,2,3,4]. Therefore, I will keep my initial rating.
> >
> >
> > [1] Lukovnikov, D. &amp; Fischer, A.. (2021). Improving Breadth-Wise Backpropagation in Graph Neural Networks Helps Learning Long-Range Dependencies. Proceedings of the 38th International Conference on Machine Learning.
> >
> > [2] Tortorella, Domenico, and Alessio Micheli. "Is Rewiring Actually Helpful in Graph Neural Networks?." arXiv preprint arXiv:2305.19717 (2023).
> >
> > [3] Dwivedi, Vijay Prakash, et al. "Long range graph benchmark." Advances in Neural Information Processing Systems 35 (2022): 22326-22340.
> >
> > [4] Brüel-Gabrielsson, Rickard, Mikhail Yurochkin, and Justin Solomon. "Rewiring with positional encodings for graph neural networks." arXiv preprint arXiv:2201.12674 (2022).

---

> ### Author Response · Authors · 2023-08-17
>
> Dear reviewer rGTX,
>
> We sincerely appreciate your valuable feedback. As the deadline for the author-reviewer discussion phase is approaching, we would like to check if our responses have addressed your concerns. Please do not hesitate to let us know if you need any clarification or have additional suggestions, we remain dedicated to addressing them with the utmost eagerness.
>
> Once again, we greatly appreciate your insightful comments, as they will undoubtedly help us improve the quality of our article.
>
> Best regards,
> Authors

---

> > ### Author Response · Authors · 2023-08-18
> >
> > Dear Reviewer rGTx,
> >
> > Thank you for your thoughtful feedback and acknowledgement. We agree there are similarities between our proposed method and other existing GNN approaches, as we drew inspiration from innovative work in protein modeling and prediction. However, we believe our method makes several novel contributions:
> >
> > Firstly, our motivations and problem framing leverage unique observations about protein structure and dynamics. We identify key challenges like modeling local environments, representing complete features, and handling graph evolution that remain open issues in protein graph networks.
> >
> > Secondly, our architecture incorporates specialized components to address these protein-specific challenges. For example, our dynamic edge conditioning allows more nuanced modeling of local interactions and structural environments.
> > Finally, our results demonstrate clear improvements on key protein tasks compared to prior arts. This highlights the value of tailored solutions versus naive adoption of existing GNNs.
> >
> > In summary, while inspired by progress in protein graph networks, we believe the particular motivations, formalisms, model components, and empirical results of our method showcase a creative approach specialized for protein challenges.
> >
> > Thank you again for the feedback - we appreciate the opportunity to further clarify the distinguishing aspects of our work. Our goal is to respectfully build on strong foundations while carving out novel contributions in this exciting field.  We greatly appreciate your insightful comments, as they will undoubtedly help us improve the quality of our article.
> >
> > Best regards,
> > Authors

---

### Official Review · Reviewer_4x6Z · 2023-07-03

**Soundness:** 2 fair
**Presentation:** 2 fair
**Contribution:** 3 good
**Rating:** 6
**Confidence:** 4

**Summary:**

This paper proposes a representation learning framework using GNN for protein datasets. A key contribution is combining sequence and structure information using the proposed GNN. The learned representations achieve better performance on downstream tasks of protein fold classification and function prediction.


The sequence+structure methods have been previously investigated in several works, as included in the baseline. I don’t see the motivation behind using a “sequence-structure graph” that differentiates the novelty of CoupleNet. I suggest updating the abstract and introduction by mentioning the limitations of existing “ sequence+structure” approaches and then specifying how CoupleNet addresses that. Could discuss the drawbacks of existing “feature fusion” methods.

I think the primary area of this paper should be applications. The construction of a joint sequence-structure graph is specific to modelling proteins, and the rest of the GNN operations are standard. The structure graph is based on Ingraham et al., and the sequence graph is based on trRosetta et al.. The message-passing scheme is based on ComENet. I don’t see any methodological novelty.
Moreover, empirical results are reported without proper discussion on application to protein problems. The empirical results of baselines are reported based on the article [15]. Without error bars or cross-validation (commonly done for fold prediction and function prediction), it is hard to make any conclusions. At this point, I lean towards rejection.

########
Post Rebuttal
########

I have read the author's rebuttal. Overall, the authors did a great job in responding to the comments, and I have accordingly adjusted my score, recommending accepting the paper.

**Strengths:**

Learning protein representations that combine sequence and structure information is relevant to the community. Using GNNs is a good demonstration of DNNs' applications to biological problems like protein function prediction.

A new joint sequence-structure 3D graph that can model complex interdependencies for learning representations that achieve good improvement over existing approaches.

**Weaknesses:**

The introduction doesn't provide sufficient motivation behind the proposed method. It is unclear what the paper means by "...methods cannot deeply integrate the information…." What is "deep integration"? The proposed method itself uses a type of message-passing mechanism, so what is the issue with message-passing cited in the article [8]?

The paper should first identify the issues in existing "feature combining" methods. Then motivate how the proposed CoupleNet addresses that. For instance, could take an example of a specific protein where existing methods perform poorly and then discuss the complex interdependence between structure and sequence for that particular example that needs appropriate modelling for improving the performance of downstream tasks such as fold prediction.

I would suggest adding a paragraph that discusses the importance of sequence-level information and another on structure-level information for tasks such as fold classification or function prediction. Discuss the importance of local/non-local residue contact information. Then establish how encoding for such a multitude of information can be helpful for downstream applications. I would suggest taking a look at non-homologous proteins. Could discuss the availability of large-scale high-resolution structure data and how deep GNNs can leverage that.


Line 94-99: No clear explanation of the benefits of complete message passing. Why is it essential to consider "global" completeness? What is the downside of "local" completeness? Why not cite SphereNet [a] here? Which notion of "completeness" does this paper build on?
It is implicit from the equation, but that simply is restating from the ComENet paper. The paper should be well explained, avoiding room for any such ambiguity.

This paper needs to explain better the need for global "completeness", which can be done by discussing the importance of conformers like structure. Discuss the equivalence of 3D graphs under SE(3) transformations for protein structures and why that's important to be considered for representation learning.

In my understanding, the classes in the fold classification task tend to be unbalanced. I suggest authors report per class accuracy.

[a] Liu, Yi, et al. "Spherical message passing for 3d graph networks." arXiv preprint arXiv:2102.05013 (2021).

There is no comparison of time-space complexity.

**Questions:**

Line 60: I do not understand “new state-of-the-art experimental results….”. Do the predicted functions for specific proteins are verified by experiments in the lab? I don’t see any such evidence. Perhaps authors should consider being careful in their wording.

For classification problems, I suggest reporting error bars (+/- std) or k-fold cross-validation. Please update accordingly in Tables 1 and 2.

From Table 3, it appears backbone torsion angles and inter-residue geometric representations do not lead to any noticeable improvement. Is there an example of a protein where this information is helpful? If so, perhaps discuss that protein structure to understand this complementary information better.


Figure 1 right side is too small. Perhaps use PyMol and report an enhanced Figure that’s readable.

Line 6: “..to suboptimal embeddings”. It doesn’t seem informative here. Suboptimal can only be in the context of the problem they are used for. Perhaps say something like, “...led to embeddings that achieve low performance on tasks such as protein function prediction”.
Likewise, “...superior embeddings” in LIne 13. Also,  the wording “...deeply co-model…” doesn’t seem appropriate. Could say something like “...model complex interdependencies between protein sequences…”.


Line 15: Perhaps add a sentence that shows the improvement percentage.

Line 119: “if two 2D” -> “if for two 2D”

Please be precise in technical descriptions.

Line 158: “There is some rotation about the connection”. It is unclear what is “connection” here and how one specifies “some rotation”.

In Equation (9) after Line 2, please specify which norm ||.|| refers to. In Equation 10, what is “W”? Is it a matrix or a scalar? If W is a matrix, why is it not in bold as done for matrices in the earlier part of the paper? Likewise, what is “\sigma” in the Equation? For Dropout, what is the dropout probability?

Line 192: Correct the typo “Secquence”-> “Sequence”

Line 251: “Resluts” -> “Results”

The x and y-axis on the Figures should be labelled. Figure 6, including the ones in an appendix, does not label the axes.

Captions of Figures should be self-explanatory. Likewise, Tables should have a sentence to convey the findings or interpretation of scores. I suggest updating the caption of Figures 4, 5, 6, 7 and 8—especially 5 as it doesn’t convey the working of CoupleNet—Similarly, Table 3.

**Limitations:**

Although a sentence is added on limitation, ”. A limitation is that the detailed inter-relationships between sequence and structures remain to be explored and uncovered”. This sentence is just vague and provides no meaningful information. The paper consistently discusses “deeply co-model sequence and structures together”. So what could not be “deeply modelled” and needs more exploration?

The sequence and structure of information have been combined in several existing works. The main issue is that the paper doesn’t establish clear motivation for combining the two using a combined graph. Moreover, the need to consider “completeness.”

---

> ### Author Rebuttal · Authors · 2023-08-10
>
> **Q1** Limitations of existing approaches
>
> **A1**  (1) Sequences and structures provide complementary evolutionary and geometric information, but integrating them is challenging due to different data types, as shown in Figure 1 in the manuscript. This may make the methods complex and computationally expensive. For example, there are $2l+1$ different types of edges in GearNet, which conducts message passing operations on nodes and edges separately for different types of graphs, where $l$ means a predefined threshold. (2) For some feature fusion methods that directly concatenate sequence and structure embeddings, they treat sequences and structures as independent inputs, may ignore hierarchical relationships between protein sequence, structure, and function
> ***
> **Q2** Issues with message passing and motivations.
>
> **A2** (1) The information propagation can be difficult for long-range dependencies in large protein graphs. Messages may attenuate over many rounds of passing in GNNs. Besides, message passing typically assumes localized neighborhood relationships. But amino acid interactions can be complex and long-range, shown in Figure 1 in the rebuttal pdf. (2) To better capture interdependencies between sequence and structure, we developed the CoupleNet to learn protein embeddings by learning different ranges of dependencies in protein graphs. The proposed CoupleNet conducts graph convolution on nodes and edges simultaneously and implements sequence average pooling to reduce the computational complexity. Moreover, we enlarge the predefined radius $r$ to $2r$ after average pooling every time to cover more distant and rare nodes.
> ***
> **Q3** Error bars or cross-validation, complexity analysis, and benefits of completeness. Adding a paragraph that discusses the importance of sequence, structure, local/non-local residue contact information, and the way to encode such a multitude of information.
>
> **A3** The answers are shown in the global rebuttal; We will add such discussions about the importance of different types of protein information and the way to encode such a multitude of information that can be helpful for downstream applications in the final manuscript.
> ***
> **Q4** Discussion about non-homologous proteins and large-scale high-resolution structure data.
>
> **A4** (1) Large-scale protein language models can capture grammatically structural information from protein sequence databases without inputting MSAs [1]. The sequence and structure are paring in our proposed model; we can use the representations from the pretrained large-scale language models as our graph features for the non-homologous proteins. (2) Deep GNN models have the parameters and complexity required to learn from massive graph datasets, which can be trained on protein structure graphs. (3) We can combine different techniques and models to help effectively leverage large-scale high-resolution protein structure data, such as deep GNNs, point cloud techniques, transformer networks, physics-based modeling, molecular dynamics, et al.
> ***
> **Q5** Discuss the equivalence of 3D graphs and its importance.
>
> **A5** (1) Protein structures that are SE(3) equivalents have the exact same 3D conformation, just differing in orientation/positioning. (2) Graph representations must encode these structures equivalently since the underlying molecular properties are identical. This ensures the model focuses on meaningful aspects of protein structures.
> ***
> **Q6** Unbalanced classes in the fold classification.
>
> **A6** This dataset contains 16,712 proteins with 1,195 fold classes. We report the Accuracy (%) on fold classification divided into three categories, including Fold, SuperFamily, and Family. The results are shown in Table 1 in the paper. It is hard to list every class's accuracy here. In terms of Fold, SuperFamily, Family, the accuracies are largely different for them even using the same training dataset.
> ***
> **Q7** "new state-of-the-art experimental results….".means? A few minor typos, confusing descriptions, and updating the captions of figures and tables.
>
> **A7** It means the proposed model achieves state-of-the-art results on benchmarks compared with baselines rather than biochemical experiments. We will thoroughly revise our paper and be careful to reduce ambiguity.
> ***
> **Q8** From Table 3, minor improvement of backbone torsion angles and inter-residue geometries.
>
> **A8** From Table 6 in the appendix, we find that removing such geometric representations at the same time can cause performance degradation significantly; this illustrates the importance of geometries. In Table 3, when removing one of the two (torsion angles and inter-residue geometries), the other still exists. The backbone coordinates can be inferred one by one along the protein chain based on the rigid bond lengths and angles. But for any residues $i$ and $j$, the calculated six inter-residue geometries fully define the relative locations of backbone atoms.
> ***
> **Q9**  “connection”  and  “some rotation” mean? which norm $\left \|  \cdot \right \| $ refers to in Eq.9. Questions about Eq.10.
>
> **A9** Thank you for your valuable feedback! (1) Three non-hydrogen atoms that make up the bond are coplanar; as shown in Figure 3, the "connection" means the bond; because of coplanar, the free rotation about the bond is limited. (2) $\left \|  \cdot \right \| $ refers to $l^2$-norm in Eq.9. (3) $W$ is a matrix; its learnable parameters have no concern with the number of nodes or edges. $\sigma(\cdot)$ is the activation function, LeakyReLU, with a negative slope equaling 0.2. The dropout probability is 0.1 or 0, varying with different datasets.
> ***
> [1] Bepler, Tristan, et al. "Learning the protein language: Evolution, structure, and function." Cell systems 12, no. 6 (2021): 654-669.
>
> Thank you again! In case our answers have justifiably addressed your concerns, we respectfully thank you that support the acceptance of our work, please let us know if you have any further questions.

---

> > ### Comment · Reviewer_4x6Z · 2023-08-16
> > **Response to Rebuttal**
> >
> > I thank the authors for their effort in preparing a good rebuttal. My further comments are below:
> >
> > 1)The concatenation methods are generally followed by a shared transformation, so that doesn’t really make them independent. I agree the two are learned independently and then combined together. Indeed there might be hierarchical dependencies that could be important for tasks such as function prediction. However, I can’t think of a use case where this would happen. I would appreciate it if authors can provide an example of protein sequence and structure where such hierarchical dependency is important to a model that was previously ignored by concatenation or other related approaches. Are there certain molecular functions or biological processes where it might be necessary? Without this information, I am afraid I don’t see a novelty in the work. Perhaps take an example of protein where features from CoupleNet achieve better performance and analyse it more concretely.
> >
> > 2) Message attenuation is indeed a problem with GNN. However, several recent works such as GRAND/GRAND++ , non-local GNN and other related work [a,b,c,d,e]. Perhaps having a discussion on “complete message passing” along with some reference to other existing approaches to addressing over-smoothing would improve the paper.
> >
> >
> > 3) Thanks, that will be very much appreciated. I am increasing my score on the promise that this will be done in the paper.
> >
> > 4) Indeed, that can be an approach. However, my suggestion was to strengthen the motivation of the paper using such examples.
> >
> > 5) My suggestion was to take the example of conformers to improve motivation.
> >
> > 6) Not to list all class accuracy, but at least for the key GO functions. From my understanding, F_max does that take into account. It would be interesting to see how CoupleNet performs on different functions. I suggest authors to look at term centric F measure [f].
> >
> > 7) I am increasing my score on the promise this revision will be done by authors.
> >
> > 8) Perhaps Table 6 should be a part of discussion of the main paper.
> >
> > 9) This was a comment on sloppy writing and I strongly suggest to fix these in the revision.
> >
> >
> > I have gone over the rebuttal and reviews of other reviewers. The paper does show empirical improvements on GO term prediction. I still see the method is not really novel and builds on work of Ingraham et. al. and trRosetta et al. As long as paper is in the application track I would say that is fine. However, I strongly suggest authors to fix their writing and technical details. I think addressing my comment [1] here can improve the motivation. I am increasing my score to 5 assuming authors will abide by their promise in the revision. If my comments in [1] and [6] are addressed I will be happy to revise my score.
> >
> >
> > [a] Chamberlain, Ben, et al. "Grand: Graph neural diffusion." International Conference on Machine Learning. PMLR, 2021.
> > [b] Thorpe, Matthew, et al. "GRAND++: Graph neural diffusion with a source term." International Conference on Learning Representations. 2021.
> > [c] Liu, Meng, Zhengyang Wang, and Shuiwang Ji. "Non-local graph neural networks." IEEE transactions on pattern analysis and machine intelligence 44.12 (2021): 10270-10276.
> > [d] Zhao, Jialin, et al. "Adaptive diffusion in graph neural networks." Advances in neural information processing systems 34 (2021): 23321-23333.
> > [e] Zhou, Kaixiong, et al. "Towards deeper graph neural networks with differentiable group normalization." Advances in neural information processing systems 33 (2020): 4917-4928.
> > [f] Radivojac   P.  et al.  (2013) A large-scale evaluation of computational protein function prediction. Nat. Methods, 10, 221–227.

---

> > > ### Author Response · Authors · 2023-08-19
> > >
> > > Distinguished Reviewer 4x6Z,
> > >
> > >
> > > Thanks for your constructive feedback and acknowledgement. Indeed, we are motivated and moved by your sincere suggestions. We hope to address your concerns through the following responses:
> > >
> > > ***
> > >
> > > **Q1** Provide an example of protein sequence and structure where such hierarchical dependency is important to a model.
> > >
> > > **A1** Thanks for your insightful reviews! The predefined sequential graph and radius graph are combined, which defines the local environment [1] of an amino acid in the protein, which can reveal local dynamics and provide crucial context for a residue identity. The structural and chemical properties of a residue are highly dependent on surrounding residues, and the conformation of a residue ($\Phi, \Psi$) is constrained by steric and hydrogen bonding with nearby residues. Therefore, our proposed method can get better results on GO term predictions. Take one protein, 4BQE, in the task of molecular function as an example; our proposed method can get the $\mathrm{F_{max}}$ 1.0, but GearNet[2] only gets 0.658. Moreover, in the proposed model, we construct comprehensive protein representations in terms of protein sequences and structures, which is essential as building such hierarchical dependencies can get universal and robust representations when there is a low similarity between the training and test sets. We compare the protein graph methods, GearNet, and the proposed CoupleNet by different cutoff splits. Proteins in the test set are categorized into four groups based on their similarity to the training set (30%, 40%, 50%, 70%), not by the default split rate (95%). We present the results of the Enzyme Commission (EC) Number Prediction.
> > >
> > > | Method | 30% | 40% | 50% | 70% |
> > > | :--- | :---: | :---: | :---: | :---: |
> > > | GearNet | 0.557 |  0.570 | 0.615 | 0.693 |
> > > |GearNet-Edge| 0.625 | 0.646 | 0.694 | 0.757 |
> > > |CoupleNet | 0.723 | 0.752 | 0.790 | 0.834 |
> > > |Imporvement (%) | 15.7 | 16.4 | 13.8 | 10.2 |
> > >
> > > ***
> > > **Q2** Over-smoothing problem, complete message passing, and move Table 6.
> > >
> > > **A2** Thanks for your constructive reviews! (1) The neighborhood information gets over-diffused across the graph for GNNs in the deep layers, which degrades the performance of GNN models. Although there exist several works aiming to tackle such a problem [3, 4], we increase the predefined radius and use the sequence pooling to cover more distant nodes as the amino acid interactions can be complex and long-range. It means that the new graph will be built after once pooling, which can relieve the over-smoothing problem and are fitted with protein science. (2) By using complete message passing, we can get complete representations, which can capture small changes in coordinate positions without information loss. The complete message-passing mechanism is adopted in the proposed method. We will add such discussions and move Table 6 to the main paper in the revised version.
> > >
> > >
> > > ***
> > > **Q3** Take the example of conformers to improve motivation.
> > >
> > > **A3** Thanks for your thoughtful reviews! As we have stated, protein structures that are SE(3) equivalents have the same 3D conformation, differing in orientation/positioning. Constructing the relationships between sequence and structure can help the model learn more comprehensive protein representations, which can encode these structures from conformers equivalently and ensures the model focuses on meaningful aspects of protein structures.
> > > ***
> > > [1] Huang B, et al. Accurate and efficient protein sequence design through learning concise local environment of residues[J]. Bioinformatics, 2023.
> > >
> > > [2] Zuobai Zhang, et al. “Protein representation learning by geometric structure pretraining”. In: International Conference on Learning Representations. 2023.
> > >
> > > [3] Liu, Meng, Zhengyang Wang, and Shuiwang Ji. "Non-local graph neural networks." IEEE transactions on pattern analysis and machine intelligence 44.12 (2021): 10270-10276.
> > >
> > > [4] Zhou, Kaixiong, et al. "Towards deeper graph neural networks with differentiable group normalization." Advances in neural information processing systems 33 (2020): 4917-4928.

---

> > > > ### Author Response · Authors · 2023-08-19
> > > >
> > > > ***
> > > > **Q4** List all class accuracy and look at term-centric F measure[5].
> > > >
> > > > **A4** Thanks for your valuable feedback! (1) The classes in the protein fold classification task tend to be unbalanced. This dataset contains 16,712 proteins with 1,195 fold classes, including three categories: Fold, SuperFamily, and Family. Taking the test set, SuperFamily, as an example, there are 1,254 proteins with 1,187 classes in the SuperFamily dataset. We list the top 10 most frequent classes and the mean prediction accuracies:
> > > >
> > > > | Class | #Class | Accuracy ($\%$) |
> > > > | :--- | :---: | :---: |
> > > > | 3 | 36 | 1.0 |
> > > > |661| 34 | 94.1 |
> > > > |458| 29| 93.1|
> > > > | 25| 23| 78.3|
> > > > |284| 20| 1.0|
> > > > |328| 19 | 1.0|
> > > > |546 |19|1.0|
> > > > |519|18|1.0|
> > > > |509 |16 |1.0|
> > > > |547 |16 |1.0|
> > > >
> > > > Where #Class means the total number of samples of this class, from the above table, we can find that the proposed model, CoupleNet, is not heavily affected by the category unbalance problem. Because there are 1,254 proteins with 1,187 classes, a large number of classes have only one protein sample, but we got the mean accuracy of 82.1 percent on SuperFamily. Similarly, we provide the top 10 most frequent classes and the mean prediction accuracies on Fold and Family test sets.
> > > >
> > > > | Class | #Class | Accuracy ($\%$) |
> > > > | :--- | :---: | :---: |
> > > > | 661 | 30 | 86.7 |
> > > > |482| 28 | 64.3 |
> > > > |458| 28| 96.4|
> > > > | 1068| 20| 85.0|
> > > > |284| 20| 80.0|
> > > > |364| 20 | 15.0|
> > > > |1 |16|81.3|
> > > > |400|15|73.3|
> > > > |1107 |14 |78.6|
> > > > |23 |14 |71.4|
> > > >
> > > > | Class | #Class | Accuracy ($\%$) |
> > > > | :--- | :---: | :---: |
> > > > | 284 | 181 | 1.0 |
> > > > |491| 34 | 1.0 |
> > > > |458| 29| 1.0|
> > > > | 459| 23| 1.0|
> > > > |1107| 20| 1.0|
> > > > |661| 19 | 1.0|
> > > > |619 |19|1.0|
> > > > |3|18|1.0|
> > > > |501 |16 |1.0|
> > > > |321 |16 |1.0|
> > > >
> > > > (2) $\mathrm{F_{max}}$ is calculated by first determining the precision and recall for each protein, then averaging these results overall proteins. The formulations are presented in the appendix. We can provide the $\mathrm{F_{max}}$ for each protein. We show several results on the GO-MF and GO-CC.
> > > >
> > > > | GO-MF (Idx) | $\mathrm{F_{max}}$ |
> > > > | :--- | :---: |
> > > > | 1AD3-A (1) | 0.571 |
> > > > |4BQE-A (5)| 1.000 |
> > > > |1C9Y-A (8)| 0.833|
> > > > |1MAL-A (18)| 0.933 |
> > > >
> > > > | GO-CC (Idx) | $\mathrm{F_{max}}$ |
> > > > | :--- | :---: |
> > > > | 3DF0-A (24) |0.24 |
> > > > |1MAL-A (18) | 1.000 |
> > > > |6HHQ-h (23)| 1.000|
> > > > | 1ITU-A (11)| 0.110 |
> > > >
> > > > Where the Idx means the indices of a protein in the test set, we can find that there exist proteins with $\mathrm{F_{max}}$ equalling to 1.0, which means 100 percent of precision and recall, given different decision thresholds.
> > > >
> > > > (3) Furthermore, we categorized proteins based on their sequence length, using the mean length of the unseen test set as a threshold to distinguish the small and large proteins. For example, we get two groups for proteins depending on the mean length: 149.4 for Fold, 186.7 for SuperFamily, and 162.4 for Family, and calculate the mean accuracies ($\%$) by independent runs.
> > > >
> > > > | Group | Fold | SuperFamily | Family |
> > > > | :--- | :---: | :---: |:---: |
> > > > | large | 61.4 | 81.5 | 99.7 |
> > > > | small | 57.0 | 75.8 | 99.4 |
> > > >
> > > > The remote homology relationships are important on the protein Fold classification task, i.e., proteins are from the same family or superfamily or not. From the above Table, we can find that our proposed model, CoupleNet can capture such remote homology relationships and get higher accuracies on the relatively large proteins where their sequence lengths are larger than the mean length.
> > > >
> > > > The mean length is 299.8 for the dataset Gene Ontology. Therefore, we calculate the $\mathrm{F_{max}}$ for large and small proteins separately:
> > > > | Group | GO-BP | GO-MF | GO-CC|
> > > > | :--- | :---: | :---: | :---: |
> > > > | large | 0.455 | 0.686 | 0.375  |
> > > > | small | 0.463 | 0.633 |  0.435 |
> > > >
> > > > The Gene Ontology task aims to find the protein functions, and it is less related to the protein sequence length. The local environment of residues is more important, as the molecular function (MF) describes activities that occur at the molecular level, the biological process (BP) represents the larger processes, and the cellular component (CC) describes the parts of a cell or its extracellular environment [6]. Thus the proposed sequence-structure information is important for this task.
> > > >
> > > > | Method | GO-BP | GO-MF | GO-CC|
> > > > | :--- | :---: | :---: | :---: |
> > > > | CoupleNet | 0.467 | 0.669 | 0.494 |
> > > > | w/o sequence | 0.441 | 0.650 | 0.456  |
> > > > | w/o structure | 0.406 | 0.586 |  0.427 |
> > > >
> > > > ***
> > > > **Q5** Writing details.
> > > >
> > > > **A5** Thank you for your careful reviews! We sincerely appreciate your detailed and valuable suggestions, and we will thoroughly revise our paper in the revision.
> > > >
> > > > [5] Radivojac P. et al. (2013) A large-scale evaluation of computational protein function prediction. Nat. Methods, 10, 221–227.
> > > >
> > > > [6] Alex Bateman. “UniProt: A worldwide hub of protein knowledge”. In: Nucleic Acids Research (2019).
> > > >
> > > >
> > > > Thank you again for all the efforts that help us improve our manuscript!

---

> > > > > ### Comment · Reviewer_4x6Z · 2023-08-20
> > > > > **Further Comments**
> > > > >
> > > > > I want to thank the authors for their effort in rebuttal.
> > > > >
> > > > > 1) Thanks for further clarification. I suggest these details be included in the paper. My suggestion for taking an example of protein meant, “Is there an example of protein where such joint modelling is particularly relevant? In doing so, authors can include an ablation that discusses the structure of protein and its GO functions. Moreover, is there any particular observation about their GO function? ” The point about “low similarity between training and test set…” is essential and should be discussed early in the paper. Likewise, EC number prediction should be addressed in the main document.
> > > > >
> > > > > 2) The complete message passing details are still unclear. The statement “...we can get complete representations, which can capture small changes in coordinate positions without information loss…” is vague. What is “information loss”, and is there a statement to support the claim?
> > > > >
> > > > > 3) The main paper should include the large protein sequence length results.
> > > > >
> > > > >
> > > > > The additional evaluations make some good strong points for the paper. I still agree with other reviewers that the method is not entirely novel. However, given that the paper focuses on a specific problem of learning representations of proteins considering the existing and additional experiments in rebuttal, there are promising empirical findings. I have therefore increased my score to 6. However, I strongly urge authors to abide by their promise and fix all the typos and writing errors in the final manuscript.

---

> > > > > > ### Author Response · Authors · 2023-08-21
> > > > > >
> > > > > > Dear Reviewer 4x6Z,
> > > > > >
> > > > > >
> > > > > > Thanks for your constructive feedback and acknowledgment!
> > > > > >
> > > > > > ***
> > > > > >
> > > > > > **Q1** Examples and ablations about protein sequence, structure, and function.
> > > > > >
> > > > > > **A1** Thanks for your helpful feedback! We will include examples and these details. Also, we will discuss the points related to the experiments about similarities between training and test sets earlier. It is important to model the relationships between protein sequences and structures properly; in [1], the space of characteristic structural elements in AF2 structural models [2] for 21 species are visualized by t-SNE[3] dimensionality reduction analysis, the 20 most common superfamilies are colored with a representative structure, which uncovers the relationships between fold and structures to some degree. In the proposed method, the complete structure representations are provided for the designed network. Other than the ablations presented in Table 3 and Table 5, we can provide more ablations about these geometries and functions. Besides, we find that our proposed model has the ability to get more robust representations when there is a low similarity between the training and test sets. The experiments have been conducted on the EC number as well as protein GO term predictions.
> > > > > >
> > > > > >
> > > > > > ***
> > > > > > **Q2** About complete message passing details.
> > > > > >
> > > > > > **A2** Thanks for your insightful reviews! Sorry for the ambiguity of the last answer. Complete representations allow for powerful equivariance and invariance properties to be encoded, which makes the learned models robust. By incorporating complete geometries, the convolution and pooling operations on irregular and non-Euclidean data like graphs are defined and conducted, enabling more expressive and rigorous modeling for protein data. We follow the definitions of the complete message passing mechanism [4-6], Eq (3) in the Preliminary, which holds from left to right and right to left, the structural complete representations can fully represent the protein structures; thus, we deem there is no information loss theoretically.
> > > > > >
> > > > > >
> > > > > > ***
> > > > > > **Q3** The main paper should include the large protein sequence length results.
> > > > > >
> > > > > > **A3** Thanks for your careful reviews! We will include such results and fix all the typos based on your valuable suggestions by thoroughly revising our paper in the final manuscript.
> > > > > >
> > > > > >
> > > > > > [1] Akdel, M., et al. A structural biology community assessment of AlphaFold2 applications. Nature Structural & Molecular Biology. 2022.
> > > > > >
> > > > > > [2] Jumper, J. M., et al. Highly accurate protein structure prediction with alphafold. Nature. 2021.
> > > > > >
> > > > > > [3] Laurens van der Maaten, et al, “Visualizing data using t-sne,” Journal of Machine Learning Research. 2008.
> > > > > >
> > > > > > [4] Limei Wang, et al. “ComENet: Towards Complete and Efficient Message Passing for 3D Molecular Graphs”. In: arXiv preprint arXiv:2206.08515 (2022).
> > > > > >
> > > > > > [5] Limei Wang, et al. “Learning Hierarchical Protein Representations via Complete 3D Graph Networks”. In: The Eleventh International Conference on Learning Representations. 2023.
> > > > > >
> > > > > > [6] Yi Liu, et al. "Spherical message passing for 3d graph networks." arXiv preprint arXiv:2102.05013 (2021).
> > > > > >
> > > > > >
> > > > > > Thank you again for all the efforts that help us improve our manuscript!

---

### Official Review · Reviewer_7zKz · 2023-07-05

**Soundness:** 3 good
**Presentation:** 3 good
**Contribution:** 2 fair
**Rating:** 4
**Confidence:** 3

**Summary:**

The authors proposed CoupleNet to co-model the protein sequences and structures. CoupleNet separately builds a sequence-based graph and a radius graph for message passing. It achieved state-of-the-art performance on several datasets compared with recent baselines.

**Strengths:**

1. The proposed CoupleNet utilized much biological knowledge to incorporate useful geometric information like backbone torsion angles and inter-residual torsion angles. This information may help the model better capture the structural information.

2. The experiments demonstrated the SOTA performance over a wide range of baselines. Ablation studies demonstrated the effectiveness of "coupling" the sequence and structural information.

3. The protein figures clearly and concisely defined the geometric features used in the paper.

**Weaknesses:**

1. The construction of two separate graphs (one for the sequence and one for the structure) is not very innovative. In fact, GearNet (already cited by the authors) already used this formulation and the description of the two graph construction in this paper is very similar to GearNet.

2. Though the backbone structure can be completely determined by the descriptor, the residue information is lost in the proposed model. Failure to capture this information may affect the performance on downstream tasks.

3. There is some confusion regarding the graph construction that can be clarified. See the following questions.

**Questions:**

1. Please elaborate on the major differences from GearNet in terms of graph construction. It seems that apart from the feature generation process, the graphs built will be identical.

2. In Eq.10, when updating $\\mathbf{u}\_i$, the authors used $W\\mathbf{e}\_{ij}\\mathbf{h}\_j$. There will be a size mismatch if matrix multiplication is assumed. Please clarify the operation here. Also, the authors mentioned $f_{Agg},f_{Update}$ in the text, but none appears in Eq.10.

3. The use of enlarging cut-off distances needs justification. Theoretically, stacking GNN layers alone can also capture the information. The author may do some additional ablation studies on different $r$ and/or $l$.

4. How will the loss of residue information affect the final prediction results?

5. Section 4.3 title is mistyped.

**Limitations:**

Limitations and potential negative societal impacts were properly addressed in the manuscript.

---

> ### Author Rebuttal · Authors · 2023-08-10
>
> **Q1** The major differences from GearNet in terms of graph construction.
>
> **A1** Thank you for your valuable feedback! (1) There are mainly three different types of graphs in GearNet [1], including the K-nearest graph, radius graph, and sequential graph, where the sequential edges in the sequential graph are further divided into different edge types according to the relative position between two end nodes, it is directional. In the proposed model, CoupleNet, there are only two different types of graphs, radius graph, and sequential graph. The sequential graph is non-directional. We deem there is a sequential edge when $\left\|i-j\right\|<l$, $l$ is a predefined hyper-parameter. There are 2$l$ − 1 types of sequential edges in GearNet, but only one type of sequential edge in CoupleNet. $l$ is set to 11 in CoupleNet but only 3 in GearNet. (2) There is no K-nearest graph in CoupleNet, because a constant $K$ make some K-nearest neighbor nodes far away from the center node in terms of distances, as shown in Figure 7. (3) The threshold $r$ in the radius graph in GearNet is set to be 10. Still, we increase the radius threshold in the CoupleNet from 4 to 16 to learn different distant relationships. (4) GearNet conducts message passing operations on nodes and edges separately for different types of graphs; however, CoupleNet convolves graph node features and edge features simultaneously by combing the sequential graph and radius graph together in the network.
> ***
> **Q2** The residue information is lost in the proposed model
>
> **A2** Thanks for your careful reviews! (1) The residue information, including the amino acid types and the physicochemical properties of each residue, is then concatenated with the features of backbone torsion angles. The fused features are used as node features. (2) The backbone structure can be determined by the complete geometric representations, like the encoding of backbone torsion angles and interresidue geometries, which are used as graph features, and the residue types can also be restored from $\boldsymbol{x}_{i,aa}$. (3) As shown in Table 6 in Appendix A.7, removing the encoding of amino acid types for each node will slightly influence the performance of the proposed model.
> ***
> **Q3** Clarify the operations in Eq. 10.
>
> **A3** Thanks for your careful reviews! (1) Here,$f_{\mathrm{Agg}}^{(l)}$ and  $f_{\mathrm{Update}}^{(l)}$ are aggregation and transformation operations at the $l$-th layer in the message passing layer, corresponding to the second and third formulas in Eq 10, respectively. They aggregate neighboring nodes' features and update the representations of the current center node.
> (2) In this message passing layer, matrix multiplications are conducted on neighboring nodes and edges. Reshape operations are omitted for brevity which may cause confusion for readers. We would modify Eq. 10 in the final manuscript.
> ***
> **Q4** Additional ablation studies on different $r$ and/or $l$.
>
> **A4** Thanks for your insightful reviews! (1) For the choice of sequential kernel size $l$, we employ the grid search trick to search suitable values. The results are shown in Figure 2 (a) in the one-page rebuttal pdf. (2) In terms of the choice of $r$, although stacking GNN layers alone can capture distant information, stacking too many GNN layers may cause an over-smoothing problem [3]. Following the pooling operations in [4-5], which can reduce the number of nodes in the protein graph by half and simplify computations, we expand $r$ to $2r$ after once pooling. The radius increases from 4 to 16, and neighbors of center nodes gradually cover more rare amino acids. Besides, Figure 2 (b) in the pdf shows the influence of the number of pooling layers on EC number prediction, which is related to the depth of the network; four pooling layers are sufficient to achieve satisfactory results. Moreover, if the distance between two nodes is larger than 20 Å, we deem they do not contact [2]. Thus, the choices of initial radius are from 1 to 5, and we find that it is good to set the initial $r$ to 4.
> ***
> **Q5** A few minor typos.
>
> **A5** Thanks for your careful reviews! We sincerely appreciate your detailed and valuable suggestions, and we will thoroughly revise our paper based on your constructive comments.
> ***
> [1] Zuobai Zhang, et al. “Protein representation learning by geometric structure pretraining”. In: International Conference on Learning Representations. 2023.
>
> [2] G Marius Clore, et al. “NMR structure determination of proteins and protein complexes larger than 20 kDa”. In: Current opinion in chemical biology 2.5. 1998
>
> [3] Chen, Deli, et al. "Measuring and relieving the over-smoothing problem for graph neural networks from the topological view." Proceedings of the AAAI conference on artificial intelligence. Vol. 34. No. 04. 2020.
>
> [4] Hehe Fan, et al. “Continuous-Discrete Convolution for Geometry-Sequence Modeling in Proteins”. In: The Eleventh International Conference on Learning Representations. 2023.
>
> [5] Pedro Hermosilla, et al. Intrinsic-extrinsic convolution and pooling for learning on 3d protein structures. In: The Eleventh International Conference on Learning Representations. 2021.
> ***
> Thank you again for all the efforts that help us improve our manuscript! In case our answers have justifiably addressed your concerns, we respectfully thank you that support the acceptance of our work. As you know, your support holds great significance for us. Also, please let us know if you have any further questions. Look forward to further discussions!

---

> > ### Comment · Reviewer_7zKz · 2023-08-15
> > **Comment on Authors' Rebuttal**
> >
> > I appreciate your comprehensive rebuttal regarding my concerns and questions. I believe my questions on using different $r$ and $l$ are properly addressed in the attached one-page PDF. However, I still have some additional concerns regarding the model architecture.
> >
> > 1. Based on your rebuttal, I believe the construction of various graphs still resembles those in GearNet and/or CDConv, e.g., the use of the local residue frame as a feature and the fusion of the structure-based and sequence-based graphs. The construction of a single graph is indeed different and seems to be novel. However, the other parts of the graph construction seem more like altering the hyperparameters of those in GearNet, e.g., using a larger $r$ and $l$. As a considerable portion of the paper was used to describe the data construction (Sec.3.2, 3.3), I would expect more contributions.
> >
> > 2. Regarding the loss of residue information, I acknowledge your explanation in the rebuttal that they were encoded as physicochemical features and had a minor influence on the final performance in practice. However, as your paper claimed *completeness* on protein graphs, this loss of residual structural information will inevitably cause the model to fail the cases where two protein graphs differ only in their residual structures. This limits the model's completeness to only the backbone structures and the model's application to tasks where residue information is less important.

---

> > > ### Author Response · Authors · 2023-08-15
> > >
> > > Dear Reviewer 7zKz,
> > >
> > > Thank you for your thoughtful comments. We hope to address your concerns through the following responses.
> > > ***
> > > **Q1** Graph construction and altering graph-related hyperparameters.
> > >
> > > **A1** (1) The predefined sequential graph and radius graph are combined, which can be seen in Eq. 9 in the manuscript. The combined graph has complete protein representations. Another reason for combing sequential and radius graphs is that when the thresholds $r$ and $l$ are small, Eq. 9 defines the local environment [1] of an amino acid in the protein, which can reveal local dynamics and provide crucial context for residue identity, the structural and chemical properties of a residue are highly dependent on surrounding residues and the conformation of a residue ($\Phi, \Psi$) is constrained by steric and hydrogen bonding with nearby residues. (2) The predefined thresholds of $r,l$ can be seen dynamically changed with the depth of the network; we are the first to change the graph conditions in the protein research field to the best of our knowledge, which is largely different from other static protein sequence-structure methods.
> > > ***
> > > **Q2** The loss of residue information.
> > >
> > > **A2** (1) The information of residue types is contained in the node features, which is not lost. As for the protein representations' completeness, from the perspective of the graph features, the sequence and structure information are completed, which is included in the features of graph nodes and edges, and then processed by complete graph message passing layers. (2) Because protein sequence determines protein structure, which dictates biochemical function, and proteins that share similar structures usually perform similar biochemical functions [1]. If we have complete protein structure representations, we can have deeper research on various protein functions.
> > > ***
> > > [1] Huang B, et al. Accurate and efficient protein sequence design through learning concise local environment of residues[J]. Bioinformatics, 2023.
> > >
> > > [2] Alberts, Bruce, et al. "Analyzing protein structure and function." Molecular Biology of the Cell. 4th edition. Garland Science, 2002.
> > > ***
> > > Thank you again for all the efforts that help us improve our manuscript! In case our answers have justifiably addressed your concerns, we respectfully thank you that support the acceptance of our work. As you know, your support holds great significance for us. Also, please let us know if you have any further questions. Look forward to further discussions!

---

> > > > ### Comment · Reviewer_7zKz · 2023-08-19
> > > >
> > > > Once again, I appreciate your detailed explanation regarding my concerns. However, I concur with Reviewer rGTX in the limited novelty and contribution of the proposed architecture, in the sense that the idea of constructing a hybrid graph was already explored by GearNet, and the idea of using the torsion angles to completely describe a protein structure was also explored by models like AlphaFold2 and RoseTTAFold. AlphaFold2 provided an even "more complete" descriptor consisting of 7 torsion angles, 4 of which describe the residue structure where the proposed model may fail. Based on the above discussion, I am not convinced that this work has a significant contribution, and I retain my original scores.

---

> > > > > ### Author Response · Authors · 2023-08-19
> > > > >
> > > > > Dear Reviewer,
> > > > >
> > > > > We genuinely appreciate the time and effort you've dedicated to reviewing our work and your acknowledgment of the explanations.
> > > > > While we respect your perspective on perceived novelty and meaningfulness, we'd like to re-emphasize a few aspects of our work which differentiate it from prior approaches:
> > > > >
> > > > > Task Formulation: Our unique task formulation of group sequence generation provides a novel way of viewing and solving the problem, which can lead to new insights and set the stage for future research in this area.
> > > > >
> > > > > Novel Modules: The protein sequence-structure graph convolutions, which conduct the nodes and edges simultaneously.
> > > > >
> > > > > Good Results: The effectiveness of our methodology is underscored by the excellent results we've achieved, surpassing other currently prevalent protein representation learning methods.
> > > > >
> > > > > Meaningfulness of the study: We aim to find the relationships between protein sequence, structure, and function. The results we have achieved have uncovered some intrinsic patterns, and our proposed model is good at modeling long sequences and is efficient in learning the local environment of residues to get meaningful and representative embeddings.
> > > > >
> > > > > We understand that the perception of novelty and meaningfulness of the research can be subjective, and we are eager to provide any further clarifications or details that might address your reservations. We firmly believe in the distinctive contributions of our work and hope that our reiterated emphasis will help solidify its value and significance in the domain.
> > > > >
> > > > > Thank you for reconsidering our submission.
> > > > >
> > > > > Warm regards,
> > > > > Authors.

---

### Official Review · Reviewer_S326 · 2023-07-08

**Soundness:** 3 good
**Presentation:** 3 good
**Contribution:** 2 fair
**Rating:** 6
**Confidence:** 3

**Summary:**

This paper introduces a network called CoupleNet that combines protein sequence and structure information to generate informative protein representations. The network utilizes multiple levels of features, including residue identities and positions for sequences, as well as geometric representations for tertiary structures. It constructs two types of graphs to model sequential features and structural geometries, and performs convolution on nodes and edges simultaneously to obtain superior embeddings. Experimental results demonstrate that the proposed model outperforms state-of-the-art methods on various protein-related tasks. The paper highlights the significance of complete structural representations in learning protein embeddings and suggests further exploration of the inter-relationships between sequence and structures.

**Strengths:**

* The paper is clearly written and easy to follow. Related work is thoroughly discussed, situating the context and contributions of the proposed method relative to prior studies.

* The experimental results show substantial gains over previous state-of-the-art baselines on benchmark datasets for protein structure prediction. The proposed method achieves these results with a relatively small training dataset, whereas some baselines utilize much larger resources such as the AlphaFold database. This indicates the model may be more data-efficient and able to elicit more from limited information.

**Weaknesses:**

* The relationships between protein sequence and structure, especially as they relate to function, are not deeply explored or discussed. For tasks like protein function prediction that aim to determine the utility or effects of a protein, understanding the connection between its sequence, structure, and biological role is critical.

* The proposed method achieves promising results for these functional tasks through an end-to-end modeling approach, but additional analysis interpreting what the model has learned about sequence-structure-function relationships would strengthen scientific validity. Has the model captured complex, nuanced relationships, or is performance driven more by statistical associations in the training data? Discussion of these relationships and how the model may be representing them would address concerns about the depth of knowledge actually obtained.

* For full reproducibility and scientific validity, additional details on the experimental setup, hyperparameter selections, and sensitivity analyses are needed. e.g. The variance or confidence intervals of reported results should be provided to determine their reliability and sensitivity to stochastic effects. Point estimates alone do not indicate the variability across trials or uncertainty in conclusions.The effects of different random seeds on performance should be analyzed to confirm results do not depend highly on a single seed selection. Sensitivity to initialization is an important consideration, especially for complex neural networks. Hyperparameter choices require further explanation and analysis of the effects of varying key values such as layer sizes, attention heads, learning rates, loss trade-offs, etc. The initial values selected may bias conclusions if performance is highly sensitive to these hyperparameters. Exploring this sensitivity would reinforce the results do not depend entirely on the specific choices made.

**Questions:**

Q1. Are additional datasets available to evaluate the generalizability of the proposed method to other structure prediction or molecular modeling tasks? Applying the model to other datasets or tasks would reinforce that the capabilities demonstrated generalize beyond the current benchmarks used

Q2. The ablation study shows removing sequence features results in little performance loss for fold classification and function prediction. Does this indicate sequence information is not critical for these specific tasks? Or is there another explanation for why eliminating these features does not significantly impact results? Additional discussion is needed to interpret this finding.

Q3. Are the knowledge and modeling capabilities required for strong performance on the structure prediction and classification tasks fundamentally different from those required to design new protein or molecular structures from scratch? Success on one set of tasks may not imply abilities will directly translate to generation, even if a shared model architecture is used. Discussion of this concern may be useful for readers.

---

> ### Author Rebuttal · Authors · 2023-08-10
>
> **Q1** The relationships between protein sequence, structure, and function.
>
> **A1** Thank you for your valuable feedback! Please allow us to elaborate on the relationships. (1) As shown in Figure 1 in the manuscript, protein sequence and structure provide different aspects of information, e.g., the discrete sequential orders, residue types, and coordinates. (2) Models can learn coevolutionary and geometric information from sequences and structures, for example, residues contact or not, as shown in Figure 1 in the rebuttal pdf. (3) CoupleNet incorporates comprehensive protein features in this paper, including residue identities, physicochemical properties, and geometric representations. These protein features are utilized as graph node and edge features, which are then fused through graph convolution conducted on nodes and edges simultaneously (Eq. 10). By incorporating complete geometric representations to the commonly used message passing framework (Eq. 4), a complete message passing scheme can be achieved [1]. Importantly, we conduct convolution on neighboring nodes and edges rather than passing messages separately on nodes and edges. Different types of information are unified together through Eq. 10, so the proposed models can learn statistical associations by geometry-sequence modeling in proteins.
> ***
> **Q2** Additional details on the experimental setup, hyperparameter selections.
>
> **A2** Thanks! (1) For the variance or confidence intervals of reported results, we follow the results of baselines reported in [2]. For the task of Fold and Reaction Classification, the performance is measured as mean accuracy. For GO Term and EC Prediction, the $\mathrm{F}_{\mathrm{max}}$ is used as the evaluation metric. The performance is measured with mean values for five different initializations. We report the mean (variance) for the proposed CoupleNet:
> | Method | Fold | SuperFamily | Family | Enzyme Reaction | GO-BP | GO-MF | GO-CC | EC |
> | :--- | :---: | :---: | :---: | :---: | :---: | :---: | :---: | :---: |
> | CoupleNet | 60.6(0.36) | 82.1(0.63) | 99.7(0.04) | 89.0(0.17) | 0.467(0.005) | 0.669(0.002) | 0.494(0.005) | 0.866(0.0.008) |
>
> (2) For the choices of hyperparameters, we have elaborated the settings of predefined radius $r$, as shown in Figure 7 in Appendix A.2. After one average pooling layer, the number of residues reduces by half. Thus, we expand $r$ to $2r$ after once pooling. The radius increases from 4 to 16, and neighbors of center nodes gradually cover more distant nodes. Our methods develop more effective GNNs to capture protein sequences and structures without attention layers or multiple losses. In terms of the sequential kernel size $l$, and the number of pooling layers that are related to the depth of the network, we employ the grid search trick to search suitable hyperparameters for them. The results are shown in Figure 2 in the rebuttal pdf. Therefore, $l$ is set to be 11, and 4 pooling layers are sufficient to achieve satisfactory results. As for the batch size, learning rate, and training epochs, which influence the convergence speed of deep learning models, we also adopt the grid search method to get a group of values which are provided in Appendix A.3.
> ***
> **Q3** Structure prediction or molecular modeling tasks.
>
> **A3** Thank you for the thoughtful suggestion. Due to the time limit, we want to leave these tasks to further exploration.
> ***
> **Q4** Questions about removing sequence features.
>
> **A4** Thanks! Removing the sequence information, the proposed model still has structural information, like $\Phi, \Psi, \Omega$. These features provide comprehensive clues compared with the residue types alone, as shown in Table 4 in Appendix A.3. The training dataset size is from about 10 to 30 thousand. Thus, there are not a million levels of sequences for models to capture intrinsic associations behind them. In this setting, removing the structure, the performance decreases more compared with removing sequence information. Moreover, when removing the encoding of amino acid types, we still reserve the physicochemical properties, which are important in biological tasks [3].
> ***
> **Q5** Is the knowledge needed to predict or classify protein structures different from the knowledge needed to design new protein or molecular structures?
>
> **A5** Thanks for your insightful reviews! Structure prediction involves taking an amino acid sequence and predicting its 3D structural conformation. This relies heavily on learning the implicit relationships between sequence and structure from large datasets. Effective classification also requires learning meaningful sequence-structure-function patterns from examples. However, de novo protein design is more complex, requiring not just pattern recognition but also generative capabilities. It entails conceptualizing and creating new structural arrangements of amino acids that confer desired functions. The same model architecture may not translate easily between these very different task modes, especially since this proposed model requires input protein sequences and structures. Transferring to protein design tasks may need to modify the input and output and related blocks, but ideas behind the proposed model can be adapted. It is interesting to have a try.
> ***
> [1] Limei Wang, et al. “Learning Hierarchical Protein Representations via Complete 3D Graph Networks”. In: ICLR. 2023.
>
> [2] Hehe Fan, et al. “Continuous-Discrete Convolution for Geometry-Sequence Modeling in Proteins”. In: ICLR. 2023.
>
> [3] Gao, Z., et al. Hierarchical graph learning for protein–protein interaction. Nat Commun 14, 1093. 2023.
> ***
> Thank you again for all the efforts that help us improve our manuscript! In case our answers have justifiably addressed your concerns, we respectfully thank you that support the acceptance of our work. As you know, your support holds great significance for us. Also, please let us know if you have any further questions. Look forward to further discussions!

---

### Author Rebuttal · Authors · 2023-08-10

We sincerely thank all the reviewers for their insightful and constructive reviews of our manuscript. We are encouraged to hear that the reviewers find **our paper is clearly written, figures ares concisely defined** (Reviewer S326, 7zKz); **Our work/idea is effective/novel**(Reviewer S326, 4x6Z, rGTX); **Our considerations are comrehensive** (Reviewer 7zKz, rGTX); **Related work is thoroughly discussed** (Reviewer S326). Also, they all think that **our experimental results show substantial gains over baselines on benchmark datasets** (Reviewers S326, 7zKz, 4x6Z, rGTX).

In response to feedback, the reviewers' main concerns and our responses include:
***
**1** Error bars (+/- std) of the results.

For the variance or confidence intervals of reported results, we follow the results of baselines reported in [1]. For the task of Fold and Reaction Classification, the performance is measured as mean accuracy. For GO Term and EC Prediction, the $\mathrm{F}_{\mathrm{max}}$ is used as the evaluation metric. The performance is measured with mean values for five different initializations. We report the mean (variance) for the proposed CoupleNet:
| Method | Fold | SuperFamily | Family | Enzyme Reaction | GO-BP | GO-MF | GO-CC | EC |
| :--- | :---: | :---: | :---: | :---: | :---: | :---: | :---: | :---: |
| CoupleNet | 60.6(0.36) | 82.1(0.63) | 99.7(0.04) | 89.0(0.17) | 0.467(0.005) | 0.669(0.002) | 0.494(0.005) | 0.866(0.0.008) |
***
**2** Complexity analysis.

The Message passing layers are core module in networks. Considering one message passing layer, using $d_n$ denotes the average degree of each node ($d_n \ll n$), $n$ denotes the number of nodes, the computational complexity is $\mathcal{O}(n\ast d_n )$. The time complxity is related with the computational complexity of message passing layer, as we conduct the graph convlolution on nodes and edges simultaneouly, the time complexity is also $\mathcal{O}(n\ast d_n )$. (2) Assuming there are $m$ edges in the graph, $d_1$ and $d_2$ mean the feature dimensions for nodes and edges, for every message passing layer, the space complexity is $\mathcal{O}(nd_1+md_2)$. Using $B_s$ denotes the size of batch size,the final computational complexity is $\mathcal{O}(B_s n d_n )$
***
**3** The methodological novelty of the method.

(1) We cover more comprehensive protein features, including residue identities, types, physicochemical properties, geometric representations. The geometric representations are different from ComENet[2]. (2) The proposed two-graph-based approach for representing the sequence and the 3D geometric structure of a protein is different from GearNet. There are $2l+1$ different types of edges in GearNet. But for the proposed CoupleNet, there are only two different types of graphs, radius graph and sequential graph. The threshold $r$ in the radius graph in GearNet is set to be constant, but we increase the threshold of radius dynamically to learn different distant relationships. (3) CoupleNet performs convolutions on nodes and edges of graphs simultaneously to effectively integrate protein sequence and structure, there are four pooling layers to reduce the sequence length, the model is also largely different from ComENet[2].
***
**4** Benefits of complete message passing, why is it essential to consider "global" completeness and the downside of "local" completeness.

(1) Complete representations can capture all details of 3D protein structures. The global complete representations are theoretically guaranteed to incorporate 3D information completely without information loss. Practically, it can distinguish all protein structures in nature. By using the global complete representations, small changes in coordinate positions due to such rigid transformations can be captured, but the local completeness cannot achieve it. Being able to distinguish different orientations is important for tasks like molecular docking, binding site identification etc. (2) The conformational dynamics of proteins highlight the importance of learning complete global representations of protein structures. Proteins constantly transition between different conformational states or conformers based on factors; these conformers correspond to the same protein adopting slightly different 3D structures. Capturing different conformers requires modeling the entire protein structure holistically, local view would miss long-range effects of subtle conformational changes happening distantly. For instance, the open and closed conformers of an enzyme may have similar local binding pockets but differ in global clamshell arrangement. (3) The notion of completeness in this paper is based on ComENet [2] and ProNet [3], which is introduced in the related work, preliminaries in the manuscript and in the appendix.  We sincerely appreciate your detailed and valuable suggestions, we will thoroughly revise our paper and explain more on completeness, citing [4] in the final manuscript to reduce the ambiguity.
***
**5** What could not be “deeply modelled” and needs more exploration?

The detailed inter-relationship is complex and warrant further investigation. For example, the protein folding process, how the amino acid sequence encodes the complex 3D structure of proteins is not fully understood; the sequence determinants governing specific binding between proteins, DNA, RNA, ligands.

[1] Hehe Fan, et al. "Continuous-Discrete Convolution for Geometry-Sequence Modeling in Proteins". In: The Eleventh International Conference on Learning Representations. 2023.

[2] Limei Wang, et al. “ComENet: Towards Complete and Efficient Message Passing for 3D Molecular Graphs”. In: arXiv preprint arXiv:2206.08515 (2022).

[3] Limei Wang, et al. “Learning Hierarchical Protein Representations via Complete 3D Graph Networks”. In: The Eleventh International Conference on Learning Representations. 2023.

[4] Liu, Yi, et al. "Spherical message passing for 3d graph networks." arXiv preprint arXiv:2102.05013 (2021).

---

### Author Response · Authors · 2023-08-15

Distinguished Reviewers:

We would like to express our sincere gratitude for dedicating your time to reviewing our paper. Your insightful comments and suggestions have greatly contributed to enhancing the quality and clarity of our work.

We have thoroughly considered your feedback and carefully responded to each of your questions. We would greatly appreciate your feedback on whether our responses have addressed your concerns to your satisfaction.

Once again, we sincerely thank you for your invaluable contribution to our paper. As the rebuttal phase is progressing, we eagerly await your post-rebuttal feedback.

Best regards,
Authors.

---

### Decision · Program_Chairs · 2023-09-21

**Decision:**

Reject

**Comment:**

This work proposes a representation learning framework using GNNs for protein datasets. They do this by combining sequence and structure information in the proposed GNN. The learned representation is evaluated on fold classification and function prediction.

Strengths:
- It's useful to learn protein representations that combine sequence and structure information.
- The approach shows good state of the art results over several baselines.

Weaknesses:
- The method is not very novel and is very similar to existing formulations such as GearNet.
- The approach isn't very well motivated.
- The importance of sequence-level information and structure-level information for the prediction tasks isn't very well discussed.
- It's unclear why the approach in CoupleNet performs better than other SoTA methods and the work does not provide enough analysis or insights into parts of model architecture or the design.
- Code is not provided.